# Structural and mutational analysis of the ribosome-arresting human XBP1u

**Vivekanandan Shanmuganathan[1†], Nina Schiller[2†], Anastasia Magoulopoulou[2], Jingdong Cheng[1], Katharina Braunger[1], Florian Cymer[2], Otto Berninghausen[1], Birgitta Beatrix[1], Kenji Kohno[3], Gunnar von Heijne[2,4]\*, Roland Beckmann[1]\***

[1]Gene Center, Department of Biochemistry, Center for integrated Protein Science Munich (CiPSM), Ludwig-Maximilians-Universität München, Munich, Germany; [2]Department of Biochemistry and Biophysics, Stockholm University, Stockholm, Sweden; [3]Institute for Research Initiatives, Nara Institute of Science and Technology, Takayama, Japan; [4]Science for Life Laboratory, Stockholm University, Solna, Sweden

**Abstract** XBP1u, a central component of the unfolded protein response (UPR), is a mammalian protein containing a functionally critical translational arrest peptide (AP). Here, we present a 3 Å cryo-EM structure of the stalled human XBP1u AP. It forms a unique turn in the ribosomal exit tunnel proximal to the peptidyl transferase center where it causes a subtle distortion, thereby explaining the temporary translational arrest induced by XBP1u. During ribosomal pausing the hydrophobic region 2 (HR2) of XBP1u is recognized by SRP, but fails to efficiently gate the Sec61 translocon. An exhaustive mutagenesis scan of the XBP1u AP revealed that only 8 out of 20 mutagenized positions are optimal; in the remaining 12 positions, we identify 55 different mutations increase the level of translational arrest. Thus, the wildtype XBP1u AP induces only an intermediate level of translational arrest, allowing efficient targeting by SRP without activating the Sec61 channel.
DOI: https://doi.org/10.7554/eLife.46267.001

**\*For correspondence:**
gunnar.von.heijne@dbb.su.se
(GH);
beckmann@genzentrum.lmu.de
(RB)

[†]These authors contributed
equally to this work

**Competing interests:** The
authors declare that no
competing interests exist.

**Reviewing editor:** Ramanujan S
Hegde, MRC Laboratory of
Molecular Biology, United
Kingdom

## Introduction

Polypeptide stretches, which can induce ribosomal stalling to regulate gene expression, are called ribosomal arrest peptides (AP). While being synthesized as a nascent chain, they traverse through the ribosomal tunnel and establish stable contacts with the tunnel wall. This eventually distorts one of the ribosomal active site, usually the peptidyl transferase center (PTC), and inhibits further peptide bond formation (*Wilson et al., 2016*). In some cases the capacity of APs to stall translation is not only dependent on the AP sequence but also requires an external co-effector molecule, such as arginine and tryptophan in the case of arginine attenuator peptide AAP (*Wang and Sachs, 1997*) and TnaC leader peptide (*Gong and Yanofsky, 2002*), to exert its function. In contrast, several bacterial APs stall independently of any additional small molecules, yet, are sensitive to pulling force on the nascent chain, thereby serving as force sensors (SecM [secretion monitor], MifM, VemP [Vibrio export monitoring peptide] [*Butkus et al., 2003*; *Chiba et al., 2009*; *Ishii et al., 2015*; *Ismail et al., 2012*]). Depending on the context APs can inhibit translation during elongation (SecM, VemP) (*Ishii et al., 2015*; *Su et al., 2017*; *Tsai et al., 2014*), termination (TnaC, CMV uORF2 (cytomegalovirus upstream open reading frame 2) and SAM-DC uORF (S-adenosyl-methionine decarboxylase) (*Gong et al., 2001*; *Janzen et al., 2002*; *Raney et al., 2002*) or both in some cases (ErmCL, MifM and AAP) (*Chiba and Ito, 2012*; *Fang et al., 2000*). Properties of these APs also vary in a way that some have a defined stall position (*Ishii et al., 2015*), while others have multiple stalling sites (*Chiba and Ito, 2012*; *Tsai et al., 2014*).

Cryo-EM structures of these APs within the ribosomal tunnel provided further structural and mechanistic insights, and also enabled visualization of their unique conformation and interactions, which explain their ability to inhibit ribosomal translation. Strikingly, apart from establishing contacts with the ribosomal tunnel mostly between the PTC and the tunnel constriction formed by the ribosomal proteins uL4 and uL22, no structural consensus of APs has been observed (*Wilson et al., 2016*). To the contrary, so far the structures range from an entirely extended conformation (*Zhang et al., 2015*) to almost entirely folded in secondary structure (*Matheisl et al., 2015*; *Su et al., 2017*). Until now, most structures characterize prokaryotic APs, with the exception of ribosomal termination stalling mediated by CMV uORF2 (*Matheisl et al., 2015*). Here, we analyzed the well-defined mammalian XBP1u AP, which plays a critical role in the unfolded protein response (UPR) pathway.

UPR represents the central cellular response mechanism that alleviates endoplasmic reticulum (ER) stress and adjusts ER activity levels (*Walter and Ron, 2011*). In mammalian cells, this pathway is mainly mediated by three transmembrane sensors that are located in the ER membrane: inositol requiring enzyme one alpha (IRE1α), activating transcription factor 6 (ATF6), and pancreatic endoplasmic reticulum kinase (PERK) (*Walter and Ron, 2011*). Of these three sensors, the evolutionarily most conserved is IRE1 (here, IRE1 denotes mammalian IRE1α and/or yeast Ire1); in lower eukaryotes such as yeast, it is the only known sensor mediating the UPR (*Mori, 2009*). Increasing levels of misfolded proteins during ER stress sequester BiP away from IRE1α, leading to formation of an active dimer (*Bertolotti et al., 2000*; *Okamura et al., 2000*) which is further activated by cluster formation (*Aragón et al., 2009*; *Credle et al., 2005*; *Kimata et al., 2007*; *Korennykh et al., 2009*; *Li et al., 2010*). In yeast, direct binding of unfolded proteins to the luminal core regions of IRE1-dimer or -oligomer is required for the activation (*Gardner and Walter, 2011*; *Kimata et al., 2007*), however, in mammals, direct binding model has been a matter of debate (*Kohno, 2010*).

The cytosolic domain of activated IRE1α then splices the *XBP1u* (X-box binding protein-1 unspliced) mRNA on the ER membrane, producing *XBP1s* (spliced) mRNA, which codes for an active transcription factor. The splicing reaction involves removal of a 26-nt intron from *XBP1u* mRNA, which leads to a translational frame-shift and the replacement of C-terminal segment in XBP1u downstream of the splicing site (*Calfon et al., 2002*; *Yoshida et al., 2001*). Once translocated to the nucleus, the XBP1s transcription factor activates genes encoding ER-resident chaperones and folding enzymes, the components of ER associated protein degradation and the proteins that function in secretory pathway, which together increase ER size and activity (*Figure 1*) (*Shaffer et al., 2004*; *Sriburi et al., 2004*).

Cytosolic *XBP1u* mRNA is recruited into the proximity of IRE1α on the ER membrane via an ingenious mechanism (*Figure 1*). XBP1u has two hydrophobic domains, HR1 and HR2, and a C-terminal AP of about 26 residues which pauses the translating ribosome when residing in the ribosomal exit tunnel (*Yanagitani et al., 2011*; *Yanagitani et al., 2009*). During this temporary pause in translation, HR2 is exposed outside the ribosome exit tunnel and can recruit the signal recognition particle (SRP) (*Kanda et al., 2016*). As a result, the paused XBP1u ribosome-nascent-chain mRNA complex (XBP1u-RNC) is targeted to the Sec61 protein-conducting channel on the ER membrane, where mRNA splicing by IRE1α is now possible (*Kanda et al., 2016*; *Plumb et al., 2015*). Given the moderate hydrophobicity of HR2, translational pausing is required for efficient recruitment of SRP by the stabilized XBP1u-RNC, and is critical for proper IRE1α-mediated UPR (*Kanda et al., 2016*; *Plumb et al., 2015*).

Here, we have used two complementary approaches, structural analysis and saturation mutagenesis, in order to decipher the structural basis and mechanism of the XBP1u AP activity. We show that the XBP1u AP makes extensive contacts with ribosomal tunnel components and forms a unique turn in close proximity to PTC in the ribosomal exit tunnel. Notably, the conformation of the XBP1u AP is unaltered within the ribosomal tunnel when the paused complex is bound to SRP or to the Sec61 complex, implying that the XBP1u AP does not function as a force-sensitive switch in the UPR pathway in vivo. By saturation mutagenesis, we observe that many but not all XBP1u residues constituting the turn are optimized for translational arrest. Finally, we identify XBP1u AP variants of increased arrest potency that may be useful as tools for in vitro force-sensing studies.

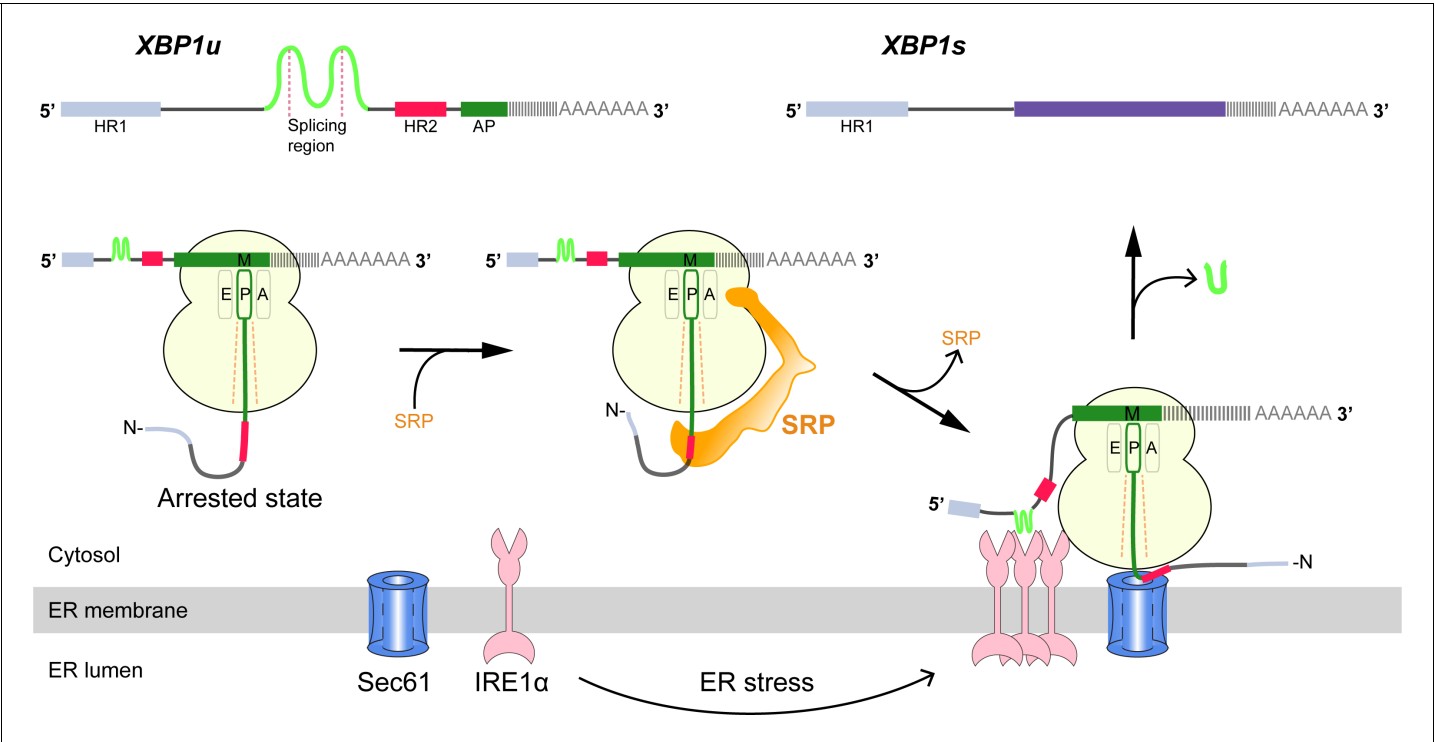

**Figure 1.** Schematic representation of the IRE1α-XBP1u pathway mediating UPR. Interaction of the XBP1u nascent chain with the ribosomal exit tunnel leads to translational pausing, resulting in SRP recruitment to the RNC, followed by targeting to Sec61 on the ER membrane. IRE1α localized near Sec61 during ER stress can splice *XBP1u* mRNA to *XBP1s* mRNA, which acts as transcription factor in alleviating ER stress.

DOI: https://doi.org/10.7554/eLife.46267.002

## Results

### Generation and cryo-EM analysis of XBP1u-paused ribosome-nascent chain complexes

We structurally characterized the paused ribosomal complex (XBP1u-RNC) by cryo-EM and single particle analysis using a mutant version of XBP1u (S255A, full length numbering), which was shown previously to pause ribosomes more efficiently than wildtype XBP1u (*Yanagitani et al., 2011*). The construct used for the RNC preparation encompassed only the HR2 domain and the XBP1u pausing sequence denoted as AP, with N- and C-terminal tags for affinity purification and detection purposes (for clarity we number the residues according to their position in the full-length protein), *Figure 2A*.

Following translation of the capped *XBP1u* mRNA in a rabbit reticulocyte lysate (RRL) in vitro translation system, paused ribosomal complexes were purified using the N-terminal His-tag on XBP1u and subjected to cryo-EM analysis. Processing of the cryo-EM dataset yielded a total of 531,952 ribosomal particles (*Figure 2—figure supplement 1*), and multiple rounds of *in silico* sorting for homogenous populations resulted in ~60% of programmed ribosomes (*Figure 2B*), with the major population of ribosomes in the non-rotated state (~42%, P- and E- site tRNA) and a minor population in the rotated, not yet fully translocated state (~18%, A/P- and P/E- site tRNA, *Figure 2C*). In both states we observed strong density for the XBP1u chain, which was connected to tRNA and extended down the ribosomal exit tunnel. The average resolutions of the paused complexes were 3.0 Å (*Figure 2—figure supplement 2A*) for the post state and 3.1 Å (*Figure 2—figure supplement 2B*) for the rotated hybrid state, respectively, with the ribosomal core reaching a resolution of 2.5 Å (*Figure 2—figure supplement 2A*). A major portion of the XBP1u peptide in the exit tunnel was resolved to between 3.0–3.5 Å for both classes (*Figure 2—figure supplement 3A,B*), whereas the resolution in the part distal to the PTC near the exit was worse than 4 Å, apparently due to flexibility of the nascent chain. We could model 24 amino acid residues of XBP1u, covering the entire AP. In both states, we observed that the ribosomes are paused with Met260 connected to

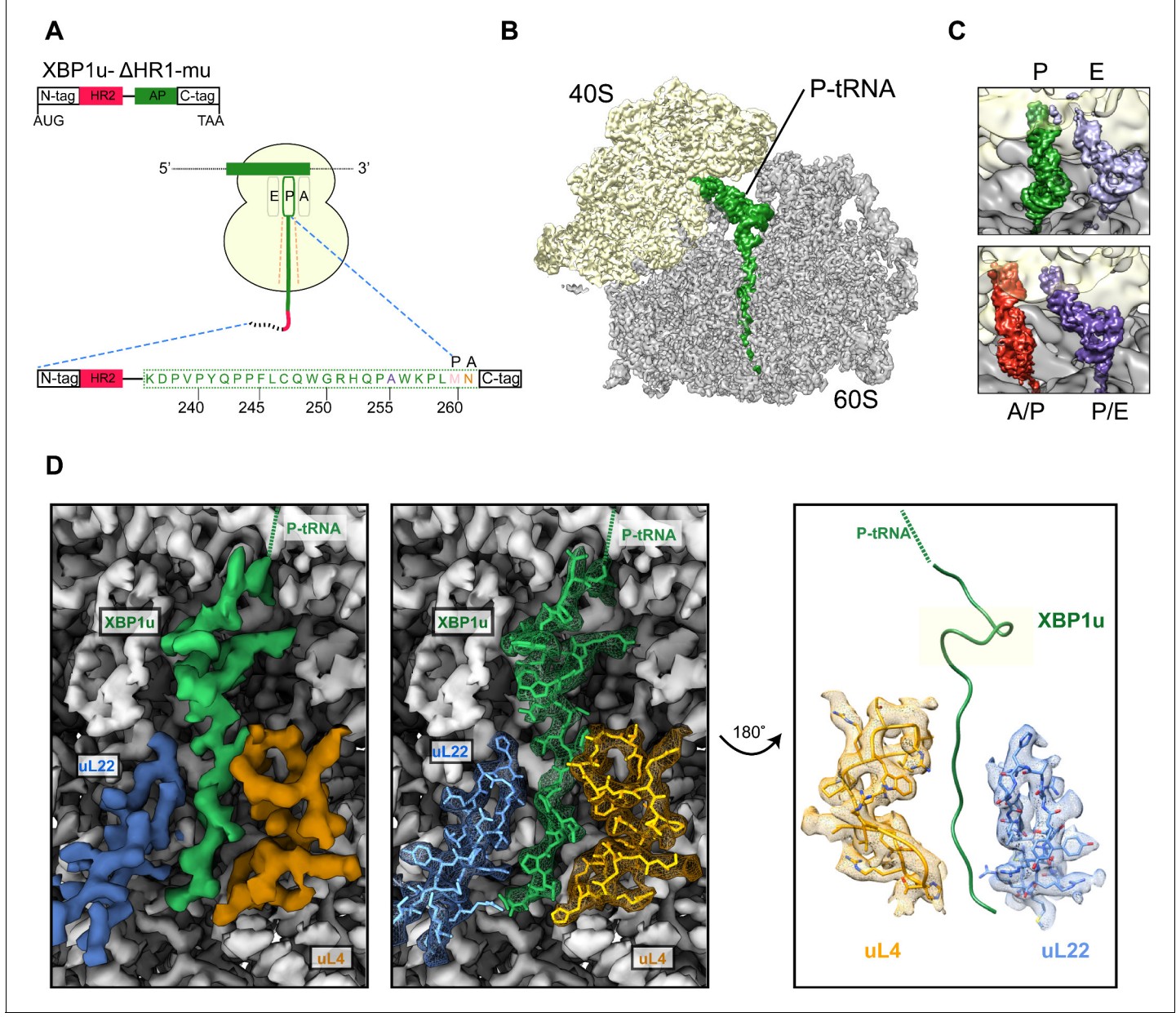

**Figure 2.** Structural analysis of XBP1u mediated ribosomal pausing. (A) Schematic representation of the XBP1u-del-HR1-mu construct used for purification. The construct encodes N-ter (8X-His, 3X-Flag tag and 3C-protease site), hydrophobic region 2 (red), AP (green) and C-ter (HA-tag). Model for nascent chain in the tunnel, and P-site and A-site positions were denoted as well. (B) Transverse section of cryo-EM structure of the paused XBP1u-RNC showing the peptidyl-tRNA (green) with small and large subunits colored in yellow and gray, respectively. Densities for nascent chain, small and large subunit are displayed at contour levels of 1.5, 1.7 and 4.1 σ, respectively. (C) Close-up views showing the two tRNA states of the XBP1u-RNC, post (top panel) and rotated (bottom panel). For the post state (top panel), P- and E-site tRNA are displayed at 3.4 and 3 σ. While small and large subunit densities are shown at 2.7 and 2.8 σ, respectively. For the rotated state (bottom panel), A/P-, P/E-tRNA, large and small subunit densities are shown at 2.6, 3.1, 3.1 and 3.3 σ, respectively. (D) Overview of the XBP1u nascent chain in the ribosomal tunnel. Surface representation of the electron density: nascent chain (green), uL4 (orange), uL22 (blue) and ribosomal tunnel (gray). Densities for nascent chain, large subunit, uL4 and uL22 are displayed at the contour levels of 2.6, 3.9, 3.2 and 4.1 σ, respectively.

DOI: https://doi.org/10.7554/eLife.46267.003

The following figure supplements are available for figure 2:

**Figure supplement 1.** Cryo-EM data processing of the XBP1u nascent chain stalled ribosomes.
DOI: https://doi.org/10.7554/eLife.46267.004

**Figure supplement 2.** Resolution of XBP1u-RNCs.
DOI: https://doi.org/10.7554/eLife.46267.005

*Figure 2 continued on next page*

*Figure 2 continued*

**Figure supplement 3.** XBP1u nascent chain resolution in the ribosomal tunnel and comparison to other known stalling peptides.
DOI: https://doi.org/10.7554/eLife.46267.006

the tRNA in the P-site, in full agreement with findings from ribosome-profiling analysis of mouse embryonic cells (*Ingolia et al., 2011*). In the following sections, we will refer to the post-state paused RNC complex for further analysis and discussion, since the nascent chain conformation is indistinguishable in both states.

## XBP1u nascent chain in the ribosomal tunnel

The majority of the visible XBP1u nascent chain adopts an extended conformation, except in the proximity of the PTC where the AP forms a prominent turn in the tunnel (*Figure 2D*). The turn is comprised of eight residues from W249 to W256, and involves the C-terminal half of the characterized XBP1u AP. Notably, the beginning of the turn is only four residues away from the PTC, suggesting that the turn within the tunnel may be critical for the pausing activity of XBP1u. Of the eight turn-forming residues, six have been previously shown to be critical for pausing by alanine scanning mutagenesis (*Yanagitani et al., 2011*), c.f., below. Interestingly, residue 255 that has been mutated from Ser to Ala in the sequence used here is part of the turn: A255 is tightly packed in the structure and the larger Ser residue may be sterically more problematic, possibly explaining why the S255A mutation makes the XBP1u a stronger AP.

The turn is located in close proximity to the PTC, above the constriction at uL22 and uL4, the narrowest portion of the tunnel. The conformation of the XBP1u peptide in the distal parts of the tunnel is similar to that observed for a non-pausing mammalian nascent chain in the mammalian ribosome (*Voorhees and Hegde, 2015*) (*Figure 2—figure supplement 3C–D*) and to other known viral and bacterial APs (CMV, MifM and VemP, *Figure 2—figure supplement 3E-G*) (*Matheisl et al., 2015*; *Sohmen et al., 2015*; *Su et al., 2017*). However, the turn observed for XBP1u is unique, and is located in a part of the tunnel where some other APs adopt α-helical secondary structure (*Figure 2—figure supplement 3E-G*).

## Interactions stabilizing the XBP1u peptide conformation

The turn in the XBP1u AP makes several key interactions with the tunnel wall and is in part in close proximity to the PTC. It is framed by two tryptophan residues (W249 and W256) and protrudes into a hydrophobic crevice in the tunnel, causing the displacement of the base G3904 (*Figure 3E*). The corresponding base in prokaryotes, A2058, constitutes, together with A2059, the so-called A-stretch in the *E. coli* ribosome which is critical for macrolide binding and drug-mediated ribosome stalling (*Wilson, 2009*). Moreover, in TnaC mediated translational arrest the hydrophobic crevice is proposed to play a critical role in recognizing free L-tryptophan in the ribosomal exit tunnel (*Martínez et al., 2014*). Therefore, it is possible that this region evolved also in eukaryotes to contribute to the sensing of nascent chains in the tunnel. The positively charged side chain of Arg251 in XBP1u forms a stabilizing salt bridge with the phosphate of A4388 (*Figure 3F*), whereas Gly250 and Gln253 engage in hydrogen bonds with A3908 and U4555, respectively (*Figure 3G,I*). Finally, Trp249 stacks internally onto Gln248 of XBP1u, and the backbone carbonyl of Arg251 makes a hydrogen bond to Lys257 within the XBP1u nascent chain (*Figure 3A and B*). Lys257 also stacks onto U4532, and this stacking interaction might influence the movement of the critical PTC base U4531 (U2585 in *E. coli*) (*Figure 3D*). Taken together, five of the eight residues that constitute the turn engage in contacts with the tunnel.

In the distal part of the tunnel, Tyr241 of XBP1u stacks with C2794 of 28S rRNA (*Figure 3H*). Three of the remaining interactions of the nascent chain in the distal tunnel region are mediated by the constriction proteins uL4 and uL22, respectively. Here, Arg71 and Ser87 of uL4, as well as Arg128 of uL22 make contacts mostly with the backbone of the nascent chain (*Figure 3J - L*).

## PTC silencing by the XBP1u peptide

Next, we asked how the unique conformation of the XBP1u peptide in the tunnel results in silencing of the peptidyl transferase activity to cause ribosomal pausing. To that end, we compared the

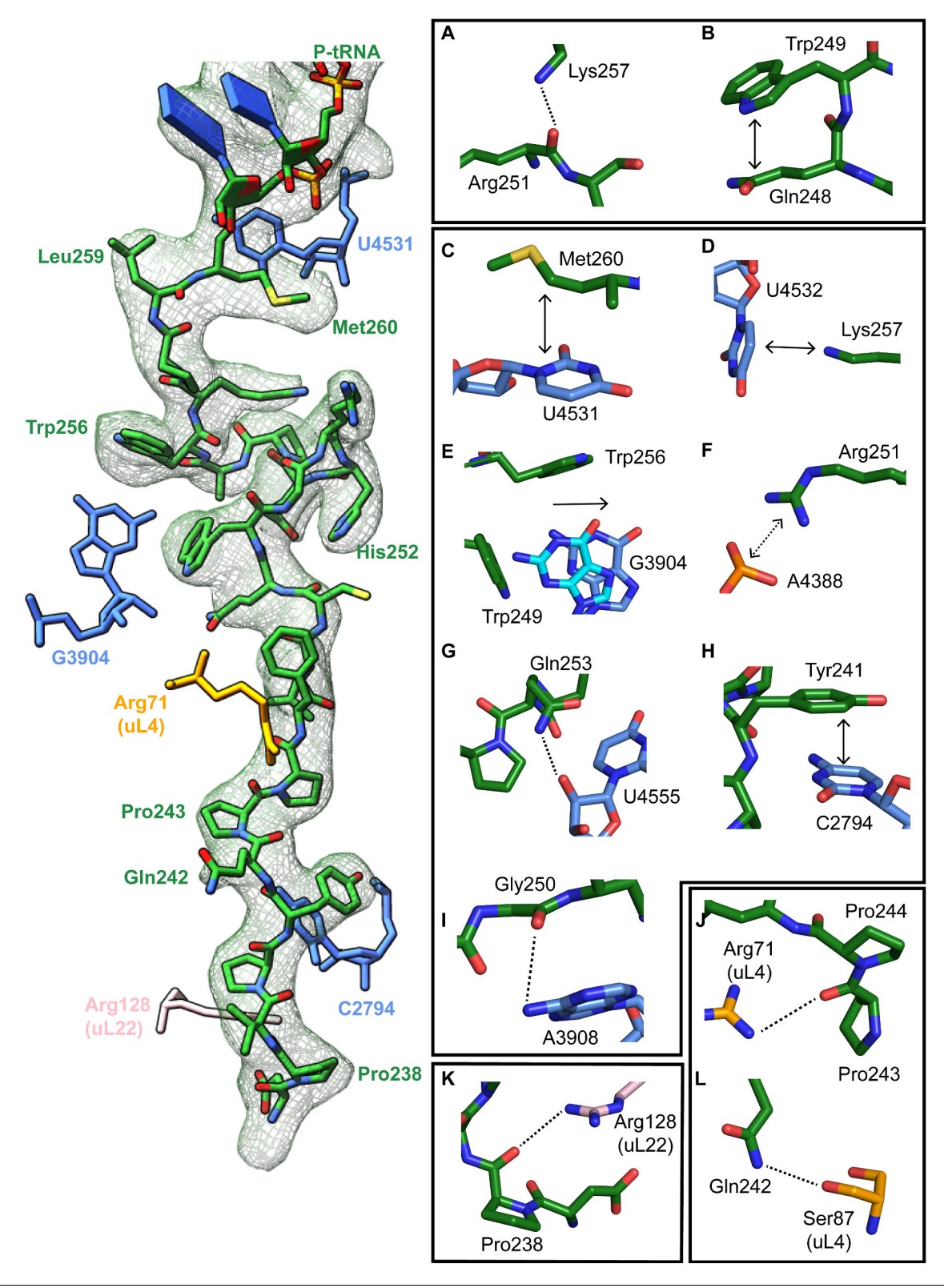

**Figure 3.** Stabilizing interactions of XBP1u nascent chain with the ribosomal exit tunnel. On the left shown nascent chain model (green) with density (gray mesh), and some interacting 28S rRNA bases and ribosomal protein residues are shown. Isolated nascent chain density is displayed at contour level of 1.28 σ. (**A**) Lys257 of XBP1u (green) is at the hydrogen bond making distance internally within XBP1u residue Arg251. (**B**) Trp249 of the XBP1u stacks internally onto Gln248. (**C**) Met260 of XBP1u makes a hydrophobic interaction with U4531 of 28S rRNA (blue). (**D**) Lys257 stacking with the base

*Figure 3 continued on next page*

*Figure 3 continued*

U4532 (**E**) Trp256 and Trp249 of XBP1u displace a ribosomal tunnel base G3904 (blue). G3904 conformation with XBP1u is compared with didemnin B treated ribosome (cyan, PDB ID 5LZS). (**F**) Arg251 of XBP1u makes a salt-bridge interaction with the exit tunnel base A4388. (**I, G**) Gly250 and Gln253 of XBP1u are in the distance for making hydrogen bond interaction with 28S rRNA bases A3908 and U4555. (**H**) Tyr241 of XBP1u stacks onto C2794. (**J–L**) Constriction site protein residues making interaction with XBP1u are shown: uL4 (orange) and uL22 (pink).

DOI: https://doi.org/10.7554/eLife.46267.007

observed PTC conformation with the available mammalian and yeast ribosome structures, either with or without accommodated A-site tRNA, respectively. Since the XBP1u-stalled RNC carries P- and E-site tRNAs but has an empty A-site, we first compared it to the reconstruction of a human 80S ribosome in the post state without A-site tRNA (*Behrmann et al., 2015*) and of a rabbit 80S ribosome in a pre-accommodation state trapped by didemnin B treatment (*Shao et al., 2016*) (*Figure 4—figure supplement 1*). Both 80S ribosomes display the classical uninduced state of the PTC before full accommodation of tRNA in the A-site, first described for bacterial ribosomes (*Schmeing et al., 2005*). It is characterized by U4531 (U2585 in *E. coli*) in a typical upward conformation, and C4398 (C2452 in *E. coli*), which is part of the so-called A-site crevice (*Hansen et al., 2003*), in the typical out-position (*Figure 4—figure supplement 1*). In contrast to some bacterial APs, we observed U4531 (U2585) in the XBP1u-stalled RNC in its canonical upward conformation. Although it interacts with the side chain of Met260 (*Figure 3C*), it appears that this base would not be hindered to switch downwards upon A-site accommodation to adopt the induced conformation. However, another nucleotide, C4398 (C2452), is in the closed conformation (*Figure 4A*, *Figure 4—figure supplement 1*), a position that under normal conditions is observed only after A-site accommodation, as in the reconstruction of the yeast 80S ribosome with A-, P-site tRNA and eIF5a (PDB 5GAK) (*Schmidt et al., 2016*). C4398 (C2452) is stabilized in the closed conformation by Leu259, which, in contrast to Met260, cannot be mutated to alanine without almost entirely loosing stalling activity (see below). Notably, this base has been implicated in A-site tRNA accommodation and peptidyl transferase activity. Therefore, the premature positioning of C4398 (C2452) in the closed conformation due to its interaction with Leu259 in an unusual conformation provides a mechanistic explanation for PTC inactivation by inhibition of A-site tRNA accommodation: Leu259 would simply

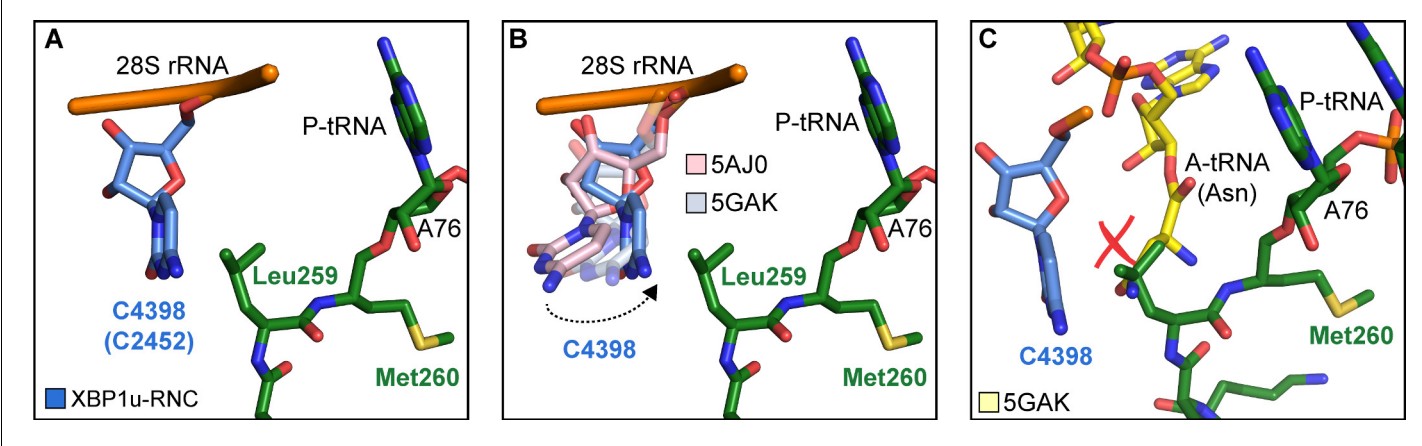

**Figure 4.** Silencing of peptidyl transferase activity by XBP1u nascent chain. (**A**) Conformation of C4398 in XBP1u-RNC (blue). (**B**) C4398 conformation in the paused complex in comparison with A-site accommodated 80S (PDB ID 5GAK, softblue) and with a post state 80S without an A-site tRNA (PDB ID 5AJ0, softpink) (**C**) Model of an incoming A-site tRNA (yellow, PDB ID 5GAK) clashes with Leu259 of XBP1u. Accommodation of A-site tRNA is prevented by XBP1u.

DOI: https://doi.org/10.7554/eLife.46267.008

The following figure supplement is available for figure 4:

**Figure supplement 1.** Comparison of C4398 (C2452) and U4531 (U2585) conformation in XBP1u-RNC with other post-state ribosome 80S models.

DOI: https://doi.org/10.7554/eLife.46267.009

clash with the incoming Asn261 tRNA (*Figure 4C*). Therefore, inhibition or delay of tRNA accommodation into the A-site appears to be the main mechanism for translational pausing by the XBP1u AP. This idea is further supported by the observation that we do not find a stable class of ribosomes in our population of stalled RNCs that carry a canonical A-site tRNA. Moreover, it can be easily imagined how pulling force applied to the nascent chain can rectify the only mildly perturbed geometry of the PTC and thereby alleviate stalling. Since we detect a substantial fraction of the RNCs in the rotated state (*Figure 2C*, *Figure 2—figure supplement 1*) we cannot exclude that the XBP1u structure may also perturb translocation as well as preventing rapid A-site tRNA accommodation. However, we do not observe any perturbation of the P-tRNA 3'-CCA end pairing with the P-loop in 28S rRNA, as was seen in other cases (*Li et al., 2019*; *Zhang et al., 2015*).

Taken together, the entire XBP1u AP contributes to pausing by interacting with the tunnel to form a unique turn structure and, facilitated by this structure, stabilizing the PTC in a conformation that disfavors A-site accommodation.

## Cryo-EM structure of XBP1u-RNC engaged with SRP and Sec61

The paused XBP1u-RNC complex has to be co-translationally targeted to and localized on Sec61 at the ER via the SRP pathway for efficient IRE1α mediated splicing of the *XBP1u* mRNA (*Kanda et al., 2016*; *Plumb et al., 2015*). Due to the AP-triggered prolonged dwell time on the ribosome, the HR2 domain of XBP1u gains sufficient affinity to be recognized by SRP. In order to analyze this special mode of SRP recruitment, and to study the state of the nascent chain within the tunnel when engaged by SRP, we generated cryo-EM structures of the paused XBP1u-RNC complex reconstituted in vitro with mammalian SRP or the Sec61 complex.

We reconstituted the purified paused XBP1u-RNC with dog SRP in vitro (see Materials and methods for details) and subjected the sample to cryo-EM analysis. After sorting for the presence of SRP and further refinement, a final reconstruction was obtained representing the paused XBP1u-RNC in the post state bound to SRP. We found the characteristic L-shaped density of SRP with its Alu-domain bound to the subunit interface connecting to the S-domain at the exit tunnel (*Figure 5A*). The final reconstruction had an average resolution of 3.7 Å (*Figure 2—figure supplement 2C*) and the SRP itself was resolved between 5–10 Å (*Figure 5—figure supplement 1D*). A recently published engaged SRP model (PDB 3JAJ) (*Voorhees and Hegde, 2015*) fits well with our observed density, and individual segments were manually inspected and fitted as rigid bodies in Coot. Analysis of the hydrophobic groove of the SRP54 M-domain, which is known to mediate the recognition and binding of canonical signal sequences, revealed a clear rod-like density resembling that of a signal sequence (*Figure 5B*). Since the only sufficiently hydrophobic peptide stretch available is HR2 of XBP1u, it is highly likely that this density indeed represents the SRP-bound HR2 domain, bound in a conformation indistinguishable from that of normal SRP-bound signal sequences. Hence, we conclude that the exposed HR2 domain on the paused XBP1u-RNC forms a helical structure upon successful SRP recruitment, which makes a canonical interaction with the M-domain of SRP54. Notably, the nascent chain density was sufficiently well resolved within the tunnel of the XBP1u-RNC-SRP complex to allow for molecular model building. At the given resolution, the conformation of the AP is identical in the presence of SRP to that of the RNC alone. The finding that SRP binding to paused XBP1u-RNCs does not lead to perturbation of the nascent chain within the tunnel strongly suggests that this state maintains the RNC in the paused state.

Next, we reconstituted the purified XBP1u-RNC complex with canine puromycin/high-salt treated rough membranes (PKRM), thereby allowing the XBP1u-RNC-Sec61 complex to form, which should represent the XBP1u-RNC after targeting to the ER. Cryo-EM analysis after solubilization with digitonin resulted in a complex paused in the post state and indeed bound to Sec61. We observed clear density for the Sec61 translocon at the tunnel exit and for the P-site tRNA-attached to nascent chain in the ribosomal tunnel. The average resolution was 3.9 Å (*Figure 2—figure supplement 2D*) and the Sec61 complex resolved to a modest resolution of around 8 Å (*Figure 5—figure supplement 1C*), due to flexibility as observed before. We performed flexible fitting of the Sec61 structure based on the position of the transmembrane segments in order to analyze the functional state of the translocon and search for additional density possibly representing the HR2 motif. When comparing with known structures of Sec61, we found that our structure represented the idle state with the lateral gate of the translocon, mainly formed by TM2 and TM7, in a closed conformation (*Figure 5C*). We could not identify any additional density for the HR2 domain of XBP1u on or near the Sec61

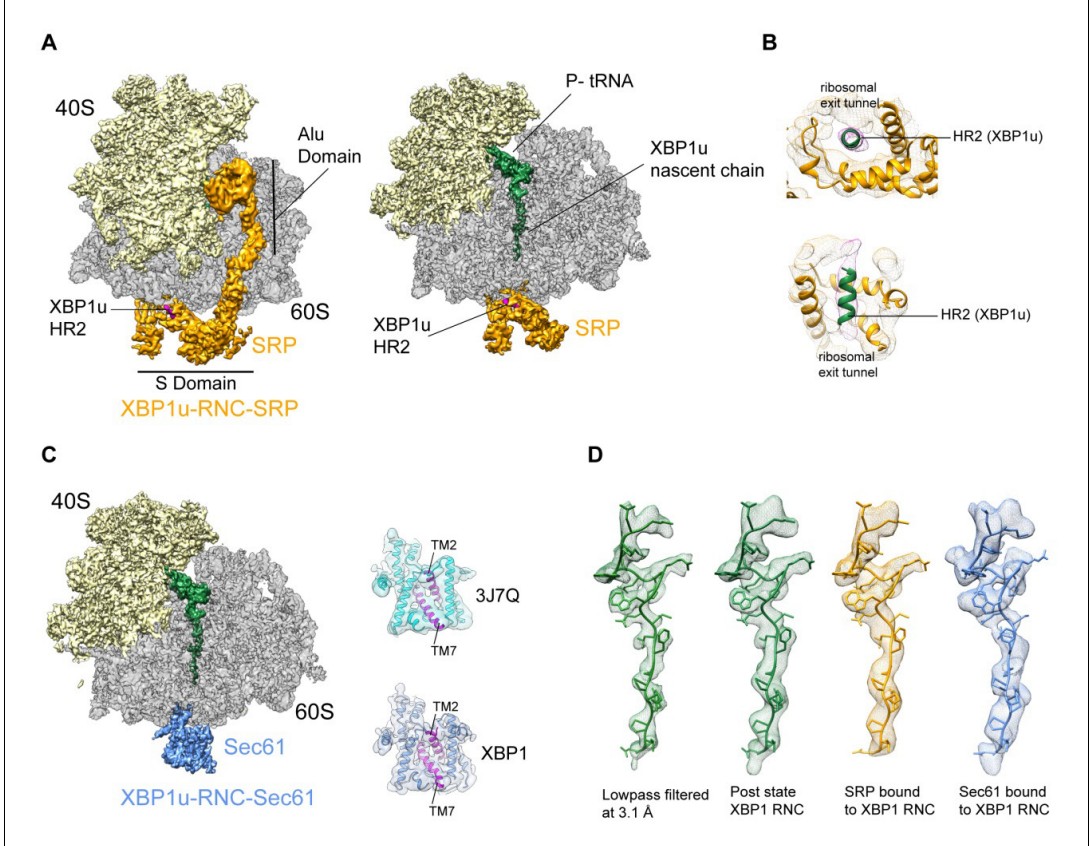

**Figure 5.** Cryo-EM structures of XBP1u-RNC with SRP and Sec61. (**A**) Cryo-EM reconstruction of XBP1u-RNC with SRP: small (yellow), large (gray), SRP (orange) and hydrophobic region 2 of XBP1u (purple). Same view, a transverse section is shown with XBP1u nascent chain and P-site tRNA (green). Densities of SRP, small and large subunit, and the nascent chain are shown at the contour levels of 2.2, 4, 3.1 and 3.1 σ, respectively. (**B**) Close-up view of SRP54 M-domain with a top and cross sectional view showing HR2 of XBP1u. (**C**) Sec61 bound to paused XBP1u-RNC. Cross sectional view: Sec61 (blue), small and large ribosomal subunits, and nascent chain density shown. Idle Sec61 model (cyan, PDB ID 3J7Q) and Sec61 model bound to XBP1u-RNC (blue). Lateral gate is highlighted in both models (purple). (**D**) Unaltered nascent chain in three different states: RNC alone (green), RNC with SRP (orange) and RNC with Sec61 (blue). Density of the nascent chain also colored respectively. From left to right: nascent chain densities are displayed at following contour levels: 1.6, 1.2, 2.7 and 1 σ, respectively.

DOI: https://doi.org/10.7554/eLife.46267.010

The following figure supplement is available for figure 5:

**Figure supplement 1.** In silico sorting and local resolution.
DOI: https://doi.org/10.7554/eLife.46267.011

complex, indicating a rather weak or transient interaction. Considering the low hydrophobicity of the HR2 domain and previous biochemical evidence that less than 10% of XBP1u becomes integrated into the ER membrane (*Plumb et al., 2015*), our data are in full agreement with the idea that HR2 can interact with, but cannot productively engage and gate, the Sec61 translocon.

The structure of the XBP1u nascent chain in the XBP1u-RNC-Sec61-complex is indistinguishable from the structures observed in the XBP1u-RNC and the XBP1u-RNC-SRP complexes, with RMSDs between the structures in the range of about 1 Å (*Figure 5D*). This finding strongly suggests that there is no change in the pausing efficiency during or after successful targeting to the ER membrane, and XBP1u is therefore unlikely to act as a force-sensitive translational switch in the UPR. Probably the long linker length of 52 amino acids between the HR2 domain and the arrest peptide prevents any potential force applied to HR2 upon interaction with SRP or Sec61 to be transmitted to the XBP1u AP. It should be noted that the above structures are obtained via in vitro reconstitution with SRP and Sec61, and also with previously described stronger stalling version (S255A) of the XBP1u AP. In our experimental conditions, without the presence of translational factors these reconstituted

complexes do not have an opportunity for further elongation even if they become transiently competent for further translational elongation on the ER membrane.

## Saturation mutagenesis of the XBP1u AP

With the structure in hand, we further characterized the XBP1u AP by saturation mutagenesis. To this end, we placed the XBP1u AP at a variable distance downstream of a hydrophobic segment (H segment) that can generate a pulling force on the nascent chain during in vitro cotranslational insertion into rough microsomal membranes (RMs) (*Ismail et al., 2012*). The construct is composed of an N-terminal part from *E. coli* leader peptidase (LepB) with two transmembrane segments (TM1, TM2), followed by a 155-residue loop, the H segment, a variable-length linker, the 25-residue long human XBP1u AP (with the S255A mutation), and a 23-residue long C-terminal tail (*Figure 6A*, *Figure 6—figure supplement 1*). An acceptor site for N-linked glycosylation located between TM2 and the H segment gets glycosylated by the luminally disposed oligosaccharyltransferase (OST) in molecules that are properly targeted and inserted into the RMs, *Figure 6A*, while non-glycosylated molecules are indicative of not properly targeted protein and therefore not subjected to pulling forces generated during membrane insertion of the H segment. Hence, only the glycosylated forms of the arrested and full-length species are used for quantitation.

When a series of constructs with varying linker-lengths is expressed in a rabbit reticulocyte lysate (RRL) in vitro translation system supplemented with RMs (*Ismail et al., 2012*), membrane insertion of the H segment is detected as a peak in a plot of the fraction of full-length protein ($f_{FL}$) against the length of the linker + AP segment ($L$, counting from residue N261; see *Supplementary file 2* for sequences), *Figure 6B* (red curve).

Based on this force profile, we chose the construct with maximal pulling force for our initial mutagenesis screen ($L$ = 43 residues, compared to $L$ = 52 residues between HR2 and the pausing site in the wildtype XBP1u).

Using the LepB-XBP1u[S255A; $L$ = 43] construct, we systematically changed each of the residues in positions 241 to 262 (position 261 corresponds to the A-site tRNA in the stalled peptide [*Ingolia et al., 2011*]) in the XBP1u AP region to all other amino acids, and measured $f_{FL}$ for each mutant. The results are summarized in *Figure 6C*. The majority of the mutations led to weaker arrest ($f_{FL} \approx$ 1.0), but a surprisingly large number of mutations reduced $f_{FL}$ from the starting value of 0.89, indicating stronger arrest variants. Particularly strong reductions in $f_{FL}$ were seen for mutations P254→[V,I,C], Q253→N, and C247→[N, K], that all have $f_{FL}$ values < 0.5.

We repeated the screen using a stronger version of the AP with the added mutation P254V from the initial screen ($f_{FL}$ = 0.46). In this second screen, we focused on positions for which mutations in the first screen gave $f_{FL} \approx$ 1, in order to detect any patterns among the mutations that weakened the efficiency of the motif. As can be seen in *Figure 6D*, all positions except A255 showed a graded response to different mutations; for the latter, all mutations gave $f_{FL}$ = 1.0 (including the back-mutation to the wildtype Ser residue). Interestingly, mutations Q253→[L, C] led to a reduction in $f_{FL}$, despite the fact that the same mutations led to an increase in $f_{FL}$ in the first screen (*Figure 6C*).

## Structural and mutagenesis hotspots in the XBP1u AP

Some general patterns are discernible from the data in *Figure 6C and D*. Eight residues in the wildtype XBP1u AP are optimal for efficient translational arrest under a strong pulling force (i.e., all mutations lead to an increase in $f_{FL}$): Y241, P244, W249 to H252, W256, and P258. Not surprisingly, the turn region W249-W256 stands out: five of the eight turn residues are optimal for translational arrest. Outside the turn region, P244 and Y241 are both located in the tunnel constriction, between uL4 and uL22. P244 makes a hydrogen bond to R71 in uL4 (*Figure 3J*) that may be weakened if the highly constrained backbone geometry of P244 is not maintained, and Y241 makes an apparently important stacking interaction with C2794 (*Figure 3H*).

In contrast, some residues in the wildtype AP are clearly sub-optimal in terms of arrest potency, and we find no less than 55 mutations in Q242, P243, F245, L246, C247, Q248, Q253, P254, S255, K257, L259, and M260 that lead to reduced $f_{FL}$ values (*Figure 6C*).

Beyond the original S255A mutation, mutations in three other key residues C247, Q253, and P254 within the AP lead to particularly strong increases in translational arrest, with $f_{FL}$ values in the range 0.2–0.4 (*Figure 6C*). Mutation of C247 to charged or polar residues increases the stalling

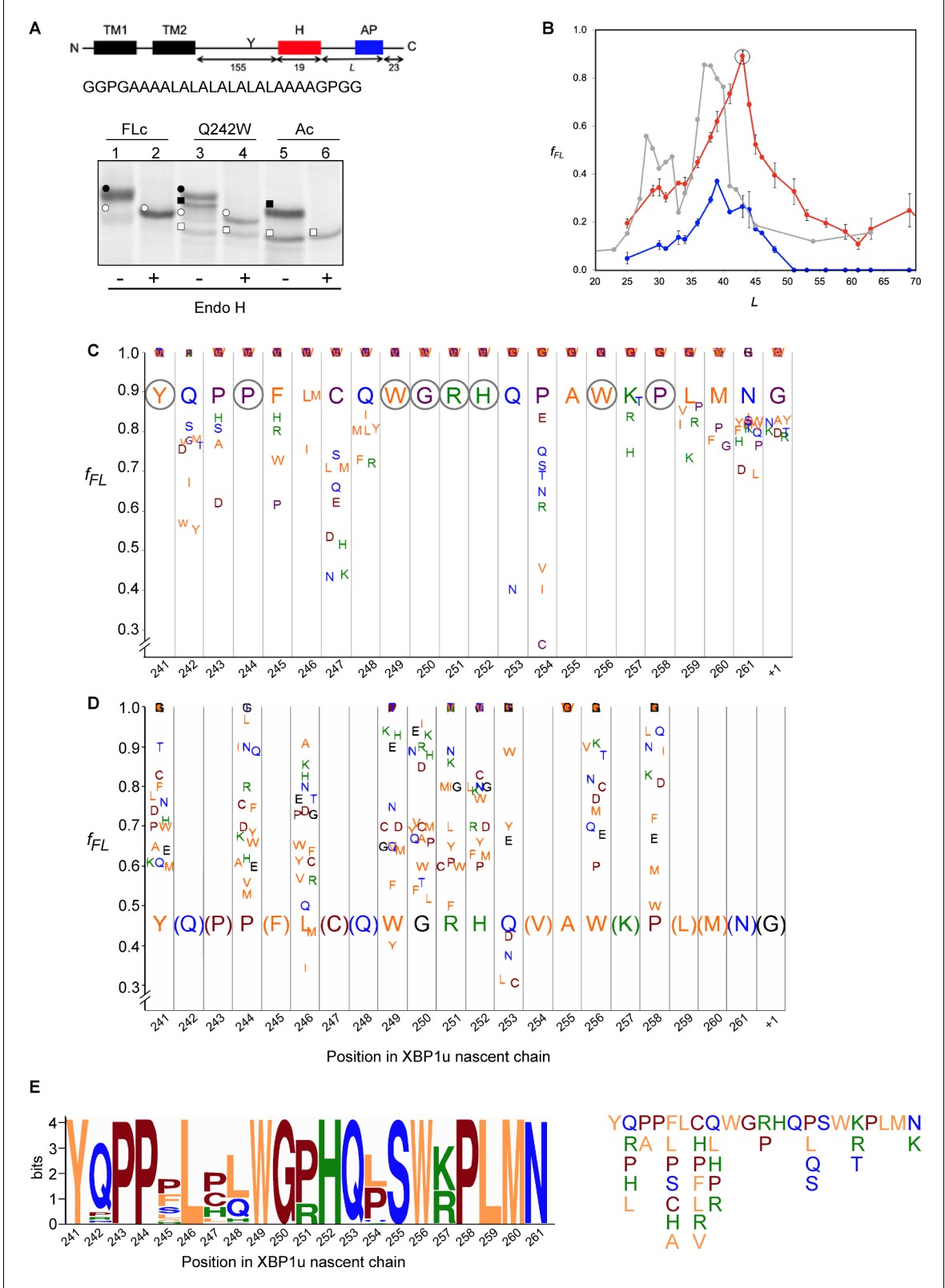

**Figure 6.** Force profile measurement and saturation mutagenesis of the XBP1u AP. (**A**) Construct used for mutagenesis screens. Y indicates the acceptor site for N-linked glycosylation. The amino acid sequence of the H segment and its flanking GGPG….GPGG residues is shown below. SDS-PAGE gel analysis of a full-length control (FLc, arrest-inactivating mutant), a construct with a Q242W mutation, and an arrest control (Ac) with stop codon immediately downstream of the AP. Full-length species are indicated by circles and arrested species by squares. Black and white colors indicate

*Figure 6 continued on next page*

*Figure 6 continued*

glycosylated and non-glycosylated species, as shown by Endo H digestion. (B) Force profiles measured for LepB-XBP1u (S255A) (red curve) and LepB-XBP1u (S255A, P254C) (blue curve) by in vitro translation in RRL supplemented with dog pancreas rough microsomes. A force profile measured in the *E. coli*-derived PURE in vitro translation system for the same construct but with the SecM(*Ms*) AP (*Ismail et al., 2012*) is included for comparison (gray curve). (C) Saturation mutagenesis of LepB-XBP1u (S255A, L = 43). Residues 241–262 were mutated to all 19 other natural amino acids and $f_{FL}$ values were determined. Residues are color-coded as follows: hydrophobic (orange), polar (blue), basic (green), acidic (brown), and G, P and C (purple). (D) Same as in C, but for LepB-XBP1u (P254V, S255A, L = 43). (E) Logo plot of the XBP1u AP, based on 90 homologous BLAST hits. The total height of each column reflects the sequence conservation in that position, and residue frequencies at a given position are indicated by the relative height of each residue in the column. The sequence variants found in the natural sequences are shown on the right.

DOI: https://doi.org/10.7554/eLife.46267.012

The following source data and figure supplements are available for figure 6:

**Source data 1.** Source for *Figure 6* force profile experiments.
DOI: https://doi.org/10.7554/eLife.46267.015
**Figure supplement 1.** LepB-XBP1u constructs.
DOI: https://doi.org/10.7554/eLife.46267.013
**Figure supplement 2.** Analysis of Cys positioning by cross-linking.
DOI: https://doi.org/10.7554/eLife.46267.014

strength. These residues presumably interact with the ribosomal tunnel by forming hydrogen bonds or salt-bridges with the phosphate backbone of rRNA, but the precise interactions cannot be easily predicted from the structure. Q253 is part of the turn, and mutating it to Asn leads to strong increase in the translational arrest. Q253 is positioned in the immediate vicinity of the extremely mutation-sensitive residue A255, and shortening the side chain by one carbon might make the turn better accommodated and more stable in the tunnel, while still allowing hydrogen bonding to U4555 (*Figure 3G*). Nine mutations in the neighboring residue P254 also increase the stalling strength, albeit to varying levels. The XBP1u turn is similar to a β-turn, but does not satisfy all the geometrical parameters and therefore is probably less stable than a canonical β-turn. Proline is not favored in β-turns, and its presence in the turn of the XBP1u nascent chain might be a result of evolution favoring weaker translational pausing instead of a highly efficient arrest.

Interestingly, mutations Q253→[L, C] led to a reduction in $f_{FL}$ (*Figure 6D*), despite the fact that the same mutations led to an increase in $f_{FL}$ in the first screen (*Figure 6C*). This is likely due to presence of Val instead of Pro in the neighboring position 254, leading an altered interaction of Q253 with the tunnel or with the nascent chain itself.

The mutagenesis data is entirely consistent with the evolutionary conservation of the XBP1u AP, as seen in the Logo plot in *Figure 6E*. None of the sequence variants in the natural XBP1u APs lead to a strong decrease in $f_{FL}$ according to *Figure 6C* (the maximal decrease is from $f_{FL} = 0.89$ to $f_{FL} = 0.52$ for C247H) and, to the extent that they have been tested, none of the sequence variants in the natural APs (except S255) lead to a strong increase in $f_{FL}$ according to *Figure 6D* (the maximal increase is from $f_{FL} = 0.46$ to $f_{FL} = 0.61$ for P244A and R251P). It is especially noteworthy that S255 is completely conserved despite the fact that the S255A mutant has much stronger arrest potency (*Figure 6D*), and that the strong Q253N mutation has not been seen so far in a natural sequence.

We conclude that, although the turn region in the XBP1u AP includes many residues that are optimal for translational pausing, the XBP1u AP is under selective pressure to maintain a rather weak translational arrest efficiency.

## Arrest-enhanced variants of the XBP1u AP can be used as force sensors

Bacterial APs have been used as force sensors to measure forces on a nascent polypeptide chain generated by cotranslational processes such as protein folding or membrane protein insertion into inner membrane (*Ismail et al., 2012*; *Nilsson et al., 2017*). To evaluate the possible use of mutant XBP1u APs in such contexts, we re-measured the force profile in *Figure 6B* using a strong XBP1u AP carrying the mutations S255→A and P254→C (blue curve in *Figure 6B*; see *Supplementary file 2* for sequences). $f_{FL}$ values are reduced throughout, while the shape of the profile persists. The P254C mutation thus reduces the APs sensitivity to both high and low pulling forces (i.e., both at the peak around L = 40–45 residues and at L ≥ 55 residues), suggesting that results obtained under high pulling forces (e.g., *Figure 6C and D*) can be extrapolated to the low-force situation that presumably

applies when the XBP1u AP performs its normal function in vivo. Because the mutant AP has a Cys residue in position 254, we considered that the enhanced arrest potency may be due to the formation of a disulfide bond with a ribosomal protein, or within the nascent chain itself. However, no crosslinked product is apparent when a gel is run under non-reducing conditions, *Figure 6—figure supplement 2A*, and $f_{FL}$ is even slightly reduced (as expected from *Figure 6C*) when the other Cys residue in the AP (C247) is mutated to Ser, *Figure 6—figure supplement 2B*.

Interestingly, the early peak at $L \approx 30$ residues seen for the same H-segment constructs expressed in *E. coli* with the SecM AP (*Ismail et al., 2012*) (gray curve, *Figure 6B*) is not clearly seen in the mammalian force profiles, suggesting that the H segment interacts differently with the Sec61 and SecYEG translocons at early stages of membrane insertion.

## Discussion

While a growing number of bacterial APs have been identified, the only well-characterized cellular mammalian arrest peptide is XBP1u, as part of the central regulator in the UPR pathway. We have determined the first high-resolution structure of a mammalian AP stalled in the ribosome exit tunnel and have carried out an extensive mutagenesis analysis, providing insights into its mode of action.

As with previously described APs, XBP1u apparently functions in a unique manner. The XBP1u AP forms a turn within the uppermost part of the tunnel to distort the PTC, thereby inhibiting translational activity. PTC bases which are known to be critical in translation elongation and termination are U4531 (U2585), U4452 (U2506) and A4548 (A2602) and, consequently, these bases are often perturbed by APs to inactivate ribosomes (*Wilson et al., 2016*). Among these, the most often distorted base is U4531 (U2585) which needs to be precisely positioned for both peptide bond formation and nascent chain release (*Schmeing et al., 2005*; *Youngman et al., 2004*). U2585 is usually either stabilized in an inactive alternative conformation or prevented from moving into the so-called induced state upon A-site tRNA accommodation (*Matheisl et al., 2015*; *Su et al., 2017*). However, in case of the XBP1u AP U4531 (U2585) appears to be neither restricted nor conformationally perturbed.

Yet, when analyzing the position of other bases in the PTC, we found not the usual ones but C4398 (C2452) in a closed state before A-site tRNA accommodation (*Figure 4*). This premature closed conformation of C4398 narrows the A-site tRNA cleft (*Gürel et al., 2009*) induced by its proximity to Leu259 of the XBP1u nascent chain, which is positioned such that it would clash with an incoming A-site Asn-tRNA. Taken together, the XBP1u nascent chain with its interactions and unique turn adopts a distinct final conformation in order to induce this mode of PTC perturbation. Therefore, compared to previously described APs, XBP1u functions similarly to some of them by preventing A-site tRNA accommodation (*Arenz et al., 2014*; *Sohmen et al., 2015*; *Su et al., 2017*), however, in a unique manner (*Figure 7*).

It was known before that mutation of critical residues involved in the described XBP1u mechanism resulted in loss of stalling activity, however, one mutation (S255A) turned out to be enhancing. Notably, our saturation mutagenesis experiments demonstrate that the XBP1u AP has not evolved to maximize its resistance to pulling forces on the nascent chain, and that many more enhancing mutations exist. P254 plays a critical role in this regard, since nine other residues in this position can all impart stronger arrest potency on the AP. The large increase in translational arrest caused by the Q253N mutation (but no other mutation in this position) points to a very specific requirement for size and hydrogen-bonding capacity in this particular position in the AP. Stronger versions of the XBP1u AP can be useful as force sensors to study cotranslational processes such as membrane-protein insertion into the ER.

We also found that the mildly hydrophobic HR2 segment of XBP1u is likely to be recognized as a canonical signal sequence by SRP, with clear density visible in the SRP54 M-domain. However, HR2 cannot engage productively with the Sec61 translocon as a signal sequence, which is consistent with previous reports of minimal membrane insertion observed for XBP1u HR2 (*Plumb et al., 2015*; *Kanda et al., 2016*). Finally, we observed that the nascent chain conformation is unaltered within the tunnel in three distinct stages of ER targeting of the ribosome-XBP1u complex: during ribosomal pausing, after recruitment of SRP and upon interaction with Sec61 translocon, arguing against the idea of a force-mediated release of translational stalling after targeting to the ER membrane. However, although we show that the conformation of the XBP1u AP is unaltered in our minimal in vitro reconstitution experiments, we need to consider the possibility of force mediated release of these

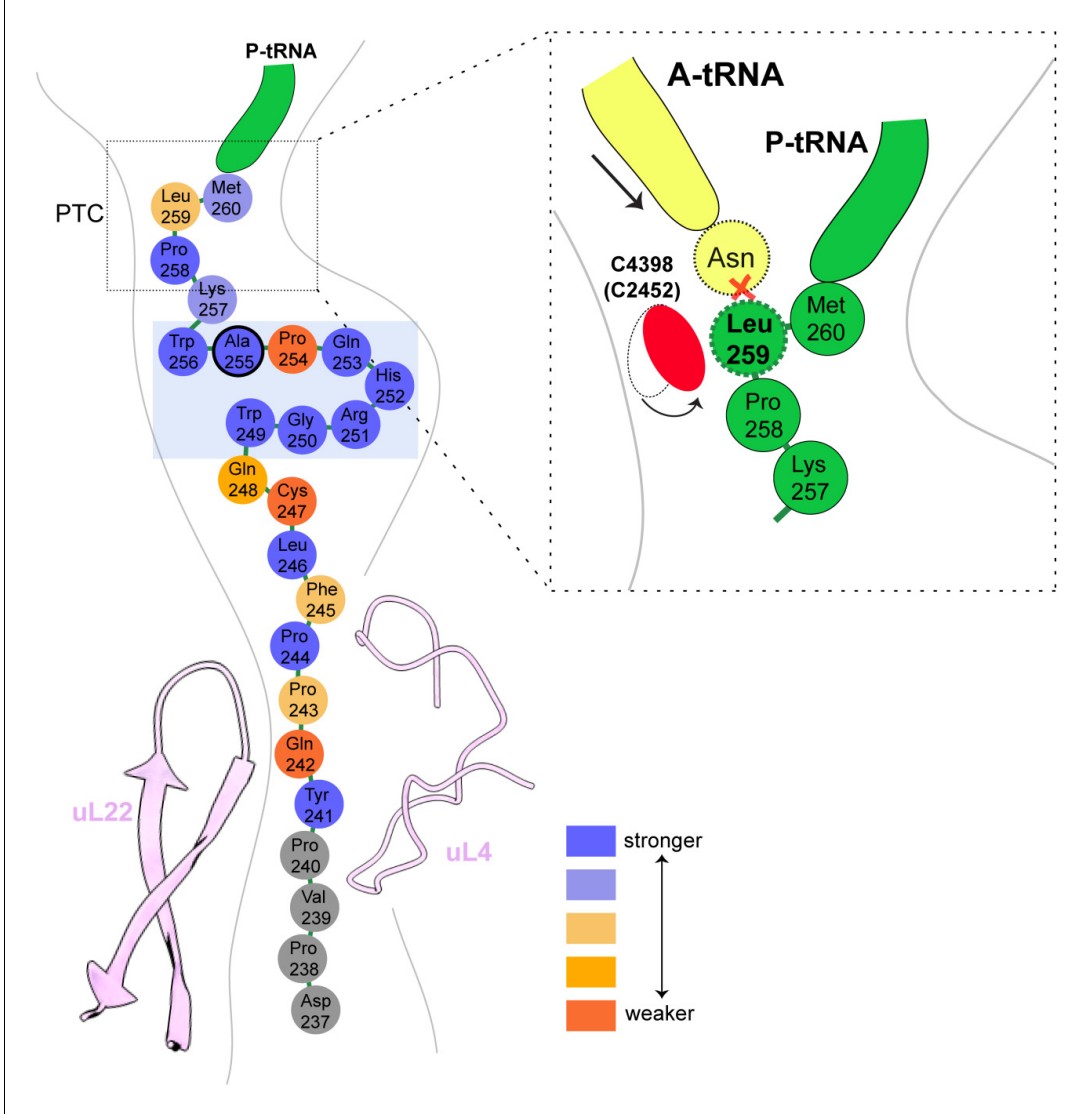

**Figure 7.** Schematic representation of the XBP1u pausing motif in the exit tunnel. XBP1u residues color coded for number residues potency based on mutagenesis data. Turn formed by XBP1u is highlighted by a light blue box. Inset shows a schematic model of the PTC summarizing the pausing mechanism.

DOI: https://doi.org/10.7554/eLife.46267.016

arrested ER targeting complexes in vivo. Previous biochemical studies and our structural evidence shows that HR2 domain of XBP1u does not engage productively with Sec61, but a subtle pulling force by the translocon might be exerted on the moderately hydrophobic HR2 domain. We cannot exclude that such a dynamic subtle pulling by the translocon, which was not possible to be visualized in our cryo-EM structures using in vitro translation conditions, might be enough to release weaker wildtype XBP1u AP in vivo.

Independently of the nature of this interaction, however, the linker length between the pausing site and the beginning of the HR2 region of XBP1u may also be responsible for uncoupling of HR2 interactions from arrest peptide conformation. From our force profile analysis, the maximal accumulation of full-length product (i.e. maximal force) occurred at a linker length of 43 amino acids, whereas in stalled XBP1u HR2 is around 52 amino acids distant from the PTC, and hence will hardly exert significant pulling force even if inserted into the ER membrane.

Based on our findings, we propose a structural and mechanistic explanation of XBP1u's role in the UPR. The XBP1u AP interaction with the ribosomal tunnel pauses ribosomes sufficiently as for

the mildly hydrophobic HR2 domain to gain competence for SRP recruitment. The recruitment of SRP ensures proper co-translational targeting, and subsequent localization of the XBP1u mRNA, to the Sec61 translocon on the ER membrane, ensuring efficient cleavage of the *XBP1u* mRNA by IRE1α. The observed unaltered states of the XBP1u nascent chain within the ribosomal tunnel suggest that neither SRP nor Sec61 release the translation stall induced by the XBP1u AP. This is consistent with the previous finding that HR2 is not hydrophobic enough for efficient membrane insertion (*Kanda et al., 2016*; *Plumb et al., 2015*).

If XBP1u-induced pausing is not released by force, we rather envision two alternative scenarios regarding the fate of the properly targeted, paused complex. First, the pausing may resolve autonomously with the given short half-life or, second, the paused complex is recognized by the Pelota/Hbs1 surveillance system as shown in yeast and recycled. The former is more likely in vivo, since the wildtype XBP1u AP is even weaker than the S255A mutant used in this study. In addition, it has also been shown biochemically that the pause is released when incubated longer during in vitro translations (*Yanagitani et al., 2011*).

In conclusion, the pausing of XBP1u might have evolved as a precise timer, which can pause ribosomes temporarily in order to allow co-translational localization of its polysome-carrying mRNA on the ER membrane for efficient splicing by IRE1α. Interestingly, the mild pausing phenotype is induced by a tight turn of the AP within the exit tunnel, and mirrored by a rather minimal perturbation of the PTC through re-positioning of just one nucleotide, C4398.

## Materials and methods

### Cloning of mutant *XBP1u*

The mutant (S255A) XBP1u (XBP1u-del-HR1-mu), which was derived from full length XBP1u (S255A) mutant as described before (*Yanagitani et al., 2011*), was then truncated to have only the HR2 domain and pausing motif with N-terminal (8X-His, 3X-Flag and 3C protease cleavage site) and C-terminal (HA-tag) for affinity purification and detection purposes. The final nucleic acid sequence of the construct used for purification:

ATGGGCCACCATCACCATCACCATCACCATGGCTCCGACTACAAGGACCATGACGGTGATTA
TAAGGATCACGACATCGACTACAAGGATGACGATGACAAGGACTACGATATCCCCACCACAC
TGGAGGTGCTCTTCCAGGGCCCTGGCGGCTCCATCTCCCCATGGATTCTGGCGGTATTGACTC
TTCAGATTCAGAGTCTGATATCCTGTTGGGCATTCTGGACAACTTGGACCCAGTCATGTTC
TTCAAATGCCCTTCCCCAGAGCCTGCCAGCCTGGAGGAGCTCCCAGAGGTCTACCCAGAAG-
GACCCAGTTCCTTACCAGCCTCCCTTTCTCTGTCAGTGGGGACGTCATCAGCCAGC
TTGGAAGCCATTAATGAACTACCCATACGATGTTCCAGATTACGCTGGATCTTAA

Corresponding final amino acid sequence of the construct:

MGHHHHHHHHGSDYKDHDGDYKDHDIDYKDDDDKDYDIPTTLEVLFQGPGGSISPWILAVLTLQI
QSLISCWAFWTTWTQSCSSNALPQSLPAWRSSQRSTQKDPVPYQPPFLCQWGRHQPAWKPLMNYPYD
VPDYAGS*

### In vitro transcription

The plasmid containing the construct was linearized with Not-I HF enzyme (NEB) at 37°C for 2 hr. mRNA for in vitro translation was prepared using the mMessage mMachine T7 Kit (Invitrogen) with linearized plasmid as the template. Capped mRNA was generated following the recommended procedures of the kit. mRNA was then extracted from the reaction mixture using lithium chloride (LiCl) precipitation.

### Rabbit reticulocyte lysate in vitro translation

Untreated crude reticulocyte lysate was purchased from Green Hectares (USA), which was then treated with Hemin and MNase, and stored at −80°C. For a 200 μl translation reaction, the 140 μl of treated lysate was used and further supplemented with 3 mM Creatine Phosphate, 30 μM yeast tRNA, 60 mM KOAc, 300 μM Mg(OAc)$_2$, 30 μM of amino-acid mixture (Promega) and 0.35 U/μl of RNAse inhibitor (SUPERase. In, Invitrogen). 80 ng of mRNA per μl of reaction volume was used for subsequent affinity purification of final XBP1u-RNC preparation.

## Purification of XBP1u- ribosome nascent chain mRNA complex (XBP1u-RNC)

mRNA was denatured by heating at 65°C for 3 min, before adding it to the translation mixture. 800 µl translation reaction mix was setup and translation was initiated with the addition of capped mRNA. Translation reaction was then incubated in 200 µl aliquots for 10 min at 37°C. Reactions were then stopped by cooling on ice, and then diluted to 2.4 ml with ice-cold buffer A (50 mM HEPES/KOH pH 7.5, 200 mM KOAc, 15 mM Mg(OAc)$_2$, 1 mM DTT, 0.1% Nikkol and 0.02 U/µl of RNAse inhibitor). Diluted reaction mix was then incubated with 800 µl of pre-equilibrated Talon metal affinity resin (Clontech, USA) at 4°C for 120 min with rotation. After incubation, beads were washed multiple times with buffer A with two intermediate washing steps with buffer A (supplemented with 10 mM imidazole). For elution of the XBP1u- ribosome nascent chain complex, the beads are then incubated with 3C protease (in buffer A) overnight at 4°C with rotation. Supernatant containing XBP1u-RNC were collected, and then centrifuged at 14,000 rpm for 10 min at 4°C to remove any large aggregates. Supernatant from this step was pelleted through 500 mM sucrose (in buffer A) cushion using TLA100.3 rotor at 90,000 rpm for 90 min at 4°C. The preparation yielded 4.2 pmol of XBP1u-RNC which was then used to make cryo-EM grid.

## In vitro reconstitution of purified XBP1u-RNC with SRP and Sec61

SRP was purified from a high salt extract of canine rough microsomes by gel filtration (Sephadex G-150), followed by ion-exchange chromatography (DEAE-Sepharose) as described before (*Martoglio and Hauser, 1998*). SRP was then further purified by sucrose centrifugation as described before (*Walter and Blobel, 1983*). XBP1u-RNC-SRP sample is prepared as follows: 1.2X molar excess of purified dog SRP was added to purified XBP1u-RNC in the presence of 2 mM GMP-PNP and 0.01% GDN (glyco-diosgenin), and incubated at 25°C for 15 min. Additional 4.5X excess of purified SRP receptor (α and β) and six-fold excess of Sec61 was added and incubated at 25°C for 15 min before being applied onto the grids for cryo-EM analysis.

Canine puromycin/high-salt treated rough membranes (PKRM) were prepared as described before (*Gogala et al., 2014*). PKRM was pre-treated with RNAsin, and were incubated with purified XBP1u-RNC for 15 min at 25°C. Membranes were then solubilized with 1.5% digitonin in Buffer A for 90 min in ice. Solubilized ribosome-translocon complexes were pelleted through sucrose cushion (with 500 mM sucrose, 0.3% digitonin, PMSF and protease inhibitor in buffer A). Pelleted complexes were resuspended in buffer A with 0.1% GDN and used for cryo-EM sample preparation.

## Cryo-electron microscopy and single particle reconstruction

XBP1u-RNC (5.2 OD$_{260}$/mL) was applied to 2 nm pre-coated Quantifoil R3/3 grids. Cryo-EM data was collected semi-automatically using EM-TOOLS acquisition software (TVIPS, Germany) on a Titan Krios TEM at a defocus range between 0.5 and 3 µm. All data were recorded on a Falcon II detector (FEI) with a nominal pixel size of 1.084 Å/pixel on the object scale. A total of 6080 micrographs were collected with a total exposure of ~28 electrons/Å$^2$ fractionated into 10 frames. All micrographs were manually inspected for ice and aggregation, and then subjected to automated particle picking with Gautomatch (https://www.mrc-lmb.cam.ac.uk/kzhang/). All classifications and refinements were performed using Relion-2.1 (Kimanius et al). Total of 531,952 ribosomal particles after 2D classification were subjected 3D classification with a prior round of 3D refinement. Initial 3D classification had two ribosomal states (post and rotated) with tRNA's. In order to further enrich the post state complex, further 3D classification was done with a mask for P-tRNA and 60S, and the resulting subsorted class with 223,773 particles were refined with a masked 60S leading to final resolution of 3 Å. The rotated state from the initial 3D classification with 94,923 particles was also refined with a 60S mask to 3.1 Å overall resolution.

A total of 10,136 micrographs were collected for XBP1u-RNC-SRP dataset and 6668 were finally subjected to automated particle picking, and further processed as mentioned above. The final subsorted class of post state-RNC with SRP was refined with a mask for 60S and SRP.

## Molecular modeling and refinement of the XBP1u-RNC

For the post state XBP1u-RNC, pdb 5LZV (*Shao et al., 2016*) was used as the initial 80S molecular model of the rabbit 80S ribosome to dock into the sharpened density. The initial fit was done with

UCSF Chimera (*Pettersen et al., 2004*), the model was further adjusted manually in Coot (*Emsley and Cowtan, 2004*) and refined using phenix.real-space_refine (*Adams et al., 2010*) with restraints obtained with the command phenix.secondary_structure_restraints. All manual adjustments for the final model were done to fit into corresponding local resolution filtered map generated with Relion 2.1 (*Kimanius et al., 2016*). Following bases of the 28S rRNA were manually inspected and adjusted in Coot: C2794, G3904, A3908, A4388, C4398, U4531 and U4532. P- and E-tRNA, mRNA was also inspected manually for proper fit into the density.

For the rotated state model, first the large subunit (60S) was fitted. For fitting the 40S, the 40S was split into two parts: the head and the body. Split small subunit models were fitted using Coot and then joined together. P/E- tRNA from the pdb 3J77 (*Svidritskiy et al., 2014*) and A/P- tRNA from pdb 3JBV (*Zhang et al., 2015*) were used as initial models in the rotated state.

Refinement for rotated state and XBP1u-RNC with SRP and Sec61 were performed as mentioned above. SRP and Sec61 models were rigid body docked and fitted in Coot, and initial models were from pdb 3JAJ (*Voorhees and Hegde, 2015*) and 6FTI (*Braunger et al., 2018*). Molprobity (*Chen et al., 2010*) was used to calculate the statistics (*Supplementary file 1*) of all the final refined models.

## Figure preparation

Figures displaying electron densities and models were generated with either UCSF Chimera (*Pettersen et al., 2004*), ChimeraX (*Goddard et al., 2018*) or Pymol (Version 1.8.2 Schrödinger, LLC). For figures (2B and 3), nascent chain density was isolated from the final sharpened map (post state), and was further Gaussian filtered at the width of 1.084 Å for figure preparation. For *Figure 2C*, tRNA's are isolated and filtered for visualization as described above. Large and small subunit densities were isolated from the Gaussian filtered (width = 4 Å) final sharpened map. For *Figure 2D* and *Figure 2—figure supplement 3A*, displayed densities were isolated from final sharpened map and further low-pass filtered at 3.1 Å. For *Figure 5A*, electron densities for SRP, small and large subunit, and the nascent chain were isolated from final sharpened map, and Gaussian filtered (width = 1.084 Å) for displaying purposes. For *Figure 5D*, nascent chain densities (left to right) are low-pass filtered at 3.1 Å, then the following three displayed nascent chain densities are Gaussian filtered at the width of 1.084 Å.

## Accession codes

The cryo-electron microscopy maps for the paused XBP1u-RNC (post and rotated states), XBP1u-RNC-SRP and XBP1u-RNC-Sec61 are deposited in the EMDataBank with the following accession codes EMD-4729, 4737, 4735 and 4745 respectively. The corresponding electron-microscopy-map based model coordinates are deposited in Protein Databank with the following accession codes: 6R5Q, 6R6P, 6R6G and 6R7Q.

## Enzymes and chemicals

Unless stated otherwise, all chemicals were from Sigma-Aldrich (St Louis, MO, USA). Oligonucleotides were purchased from MWG Biotech AG (Ebersberg, Germany). Pfu Turbo DNA polymerase was purchased from Agilent Technologies. All other enzymes were from Fermentas. The plasmid pGEM-1 and the TNT SP6 Transcription/Translation System were from Promega. [$^{35}$S]Met was from PerkinElmer.

## Construction of mutant library

Site-specific mutagenesis was performed using the QuikChange Site-Directed Mutagenesis Kit from Stratagene. All mutants were confirmed by sequencing of plasmid DNA at Eurofins MWG Operon (Ebersberg, Germany).

## Expression in vitro

Constructs cloned in pGEM-1 were transcribed and translated in the TNT Quick coupled transcription/translation system. 1 µg of DNA template, 1 µl of [$^{35}$S]-Met (10 µCi; 1 Ci1/437 GBq), 3 µl of zinc acetate dihydrate (5 µM) were mixed with 10 µl of TNT lysate mix, and samples were incubated for 30 min at 30°C. The sample was mixed with 1 µl of RNase I (Affymetrix; 2 mg/ml) and SDS sample

buffer and incubated at 30°C for 15 min before loading on a 10% SDS/polyacrylamide gel. Protein bands were visualized in a Fuji FLA-3000 phosphoimager (Fujifilm,Tokyo, Japan). The Image Gauge V 4.23 software (Fujifilm) was used to generate a two-dimensional intensity profile of each gel lane, and the multi-Gaussian fit program from the Qtiplot software package (www.qtiplot.ro) was used to calculate the peak areas of the protein bands. The fraction full-length protein ($f_{FL}$) was calculated as $f_{FL} = I_{FL}/(I_{FL} + I_A)$, where $I_{FL}$ is the intensity of the band corresponding to the full-length protein, and $I_A$ is the intensity of the band corresponding to the arrested form of the protein. Experiments were repeated three times, and SEMs were calculated.

## Acknowledgements

This work was supported by grants from the Knut and Alice Wallenberg Foundation (2012.0282), the Swedish Research Council (621-2014-3713), and the Swedish Cancer Foundation (15 0888) to GvH, and JSPS KAKENHI JP26116006 to KK. VS was supported by a DFG fellowship through the QBM (Quantitative Biosciences Munich) graduate school. This work was supported by the German Research Council (GRK1721 to RB). We also acknowledge the support of a PhD fellowship from the Boehringer Ingelheim Fonds (to KB).

## Additional information

### Funding

| Funder | Grant reference number | Author |
|---|---|---|
| Deutsche Forschungsge-meinschaft | QBM (Quantitative Biosciences Munich) Graduate School Fellowship | Vivekanandan Shanmuganathan |
| Knut och Alice Wallenbergs Stiftelse | 2012.0282 | Gunnar von Heijne |
| Vetenskapsrådet | 621-2014-3713 | Gunnar von Heijne |
| Cancerfonden | 15 0888 | Gunnar von Heijne |
| Japan Society for the Promotion of Science | JP26116006 | Kenji Kohno |
| Deutsche Forschungsge-meinschaft | GRK1721 | Roland Beckmann |
| Boehringer Ingelheim Fonds | Graduate Fellowship | Katharina Braunger |

The funders had no role in study design, data collection and interpretation, or the decision to submit the work for publication.

### Author contributions

Vivekanandan Shanmuganathan, Data curation, Formal analysis, Investigation, Methodology, Writing—original draft, Writing—review and editing; Nina Schiller, Investigation, Methodology, Writing—original draft; Anastasia Magoulopoulou, Florian Cymer, Investigation, Methodology; Jingdong Cheng, Validation, Investigation; Katharina Braunger, Resources, Methodology; Otto Berninghausen, Methodology, Cryo-EM data collection; Birgitta Beatrix, Resources, Investigation, Methodology; Kenji Kohno, Conceptualization, Resources, Writing—review and editing; Gunnar von Heijne, Conceptualization, Supervision, Funding acquisition, Writing—review and editing; Roland Beckmann, Conceptualization, Formal analysis, Supervision, Funding acquisition, Methodology, Project administration, Writing—review and editing

### Author ORCIDs

Katharina Braunger (iD) http://orcid.org/0000-0002-9067-2155
Kenji Kohno (iD) http://orcid.org/0000-0002-3503-6551
Gunnar von Heijne (iD) https://orcid.org/0000-0002-4490-8569
Roland Beckmann (iD) https://orcid.org/0000-0003-4291-3898

**Decision letter and Author response**

Decision letter https://doi.org/10.7554/eLife.46267.037

Author response https://doi.org/10.7554/eLife.46267.038

## Additional files

### Supplementary files

• Supplementary file 1. Cryo-EM data collection, refinement and validation statistics. Summary of parameters related cryo-EM data collection and processing.

DOI: https://doi.org/10.7554/eLife.46267.017

• Supplementary file 2. List of aminoacid sequences representing the constructs used in *Figure 6B*. Representing red curve (mutation S255A in the XBP1u AP) and blue curve (mutations S255A and P254C in the XBP1u AP) in *Figure 6B*.

DOI: https://doi.org/10.7554/eLife.46267.018

• Transparent reporting form

DOI: https://doi.org/10.7554/eLife.46267.019

### Data availability

Generated models have been deposited in Protein Data Bank (PDB), and accessible via the following accession codes: 6R5Q, 6R6P, 6R6G and 6R7Q. Corresponding maps have been deposited in Electron Microscopy Data Bank (EMDB) and accessible with the following accession codes: 4729, 4737, 4735 and 4745.

The following datasets were generated:

| Author(s) | Year | Dataset title | Dataset URL | Database and Identifier |
|---|---|---|---|---|
| Vivekanandan Shanmuganathan, Jingdong Cheng, Katharina Braunger, Otto Berninghausen, Birgitta Beatrix, Roland Beckmann | 2019 | Structure of XBP1u-paused ribosome nascent chain complex with Sec61. | http://www.ebi.ac.uk/pdbe/entry/emdb/EMD-4745 | Electron Microscopy Data Bank, EMD-4745 |
| Shanmuganathan V, Cheng J, Katharina Braunger, Otto Berninghausen, Birgitta Beatrix, Roland Beckmann | 2019 | Structure of XBP1u-paused ribosome nascent chain complex with Sec61. | http://www.rcsb.org/structure/6R7Q | Protein Data Bank, 6R7Q |
| Shanmuganathan V, Cheng J, Katharina Braunger, Otto Berninghausen, Birgitta Beatrix, Roland Beckmann | 2019 | Structure of XBP1u-paused ribosome nascent chain complex with SRP. | http://www.rcsb.org/structure/6R6G | Protein Data Bank, 6R6G |
| Shanmuganathan V, Cheng J, Katharina Braunger, Otto Berninghausen, Roland Beckmann | 2019 | Structure of XBP1u-paused ribosome nascent chain complex with SRP. | http://www.ebi.ac.uk/pdbe/entry/emdb/EMD-4735 | Electron Microscopy Data Bank, EMD-4735 |
| Shanmuganathan V, Cheng J, Berninghausen O, Beckmann R | 2019 | Structure of XBP1u-paused ribosome nascent chain complex (rotated state) | http://www.rcsb.org/structure/6R6P | Protein Data Bank, 6R6P |
| Shanmuganathan V, Cheng J, Berninghausen O, Beckmann R | 2019 | Structure of XBP1u-paused ribosome nascent chain complex (post-state) | http://www.rcsb.org/structure/6R5Q | Protein Data Bank, 6R5Q |
| Shanmuganathan V, Cheng J, Berninghausen O, Beck- | 2019 | Structure of XBP1u-paused ribosome nascent chain complex (post-state) | http://www.ebi.ac.uk/pdbe/entry/emdb/EMD-4729 | Electron Microscopy Data Bank, EMD-4729 |

| mann R | | | | |
|---|---|---|---|---|
| Shanmuganathan V, Cheng J, Berninghausen O, Beckmann R | 2019 | Structure of XBP1u-paused ribosome nascent chain complex (rotated state) | http://www.ebi.ac.uk/pdbe/entry/emdb/EMD-4737 | Electron Microscopy Data Bank, EMD-4737 |

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
