## [Decision Letter]

Thank you for submitting your article "Structural and mutational analysis of the ribosome- arresting human XBP1u" for consideration by *eLife*. Your article has been reviewed by four peer reviewers, and the evaluation has been overseen by Ramanujan Hegde as the Reviewing Editor and David Ron as the Senior Editor. The following individual involved in review of your submission has agreed to reveal their identity: Alexander Mankin (Reviewer #3).

The reviewers have discussed the reviews with one another and the Reviewing Editor has drafted this decision to help you prepare a revised submission.

The authors probe the molecular mechanism of translation stalling induced by the protein XBP1u, presently the only well-defined eukaryotic protein with an arrest peptide (AP) with a known function. The authors use cryo-EM to determine high-resolution structures of XBP1u-stalled ribosome nascent chain (RNC) complexes at three different steps: paused but not engaged with SRP, paused then subsequently assembled with SRP, and paused then subsequently assembled with the Sec61 translocon. The authors also use saturation mutagenesis of the AP and a system to detect force generation to test the potency of the XBP1u AP. This systematic mutagenesis verifies the key residues for stalling and identifies a palatte of variant APs with both stronger and weaker stalling propensity, which are likely to serve as useful tools in future force measurement studies. The two major conclusions of the paper are that the XBP1u AP adopts a turn structure in the ribosome exit tunnel with well-defined geometry that does not seem to be perturbed substantially by later events in the Sec61 targeting reaction. SRP binding to the hydrophobic stretch is indistinguishable from other SRP structures (at the reported resolution), and the XBP1u sequence is unable to stably engage the Sec61 pore. All of these results are consistent with XBP1u AP-induced pausing serving a kinetic role to enable XBP1u to be targeted to the Sec61 translocon. It's important to note that the authors used a "stabilizing" mutation in the AP, further emphasizing that the natural context is likely to be transient.

Overall, this is very interesting paper, the quality of the data is high, and it is generally well written. The main issues raised by the reviewers include over-interpretation of some of the findings, the need for a more complete description of some of the figures, and improved details in parts of the Materials and methods. All of these issues can be addressed without additional experimental work. The following issues should be addressed in preparing a revised manuscript:

1) The authors' claims that SRP and Sec61 do not participate in relieving pausing may lack strong experimental support and should therefore be qualified. In essence, they are looking at an artificial situation in which a purified stalled complex is mixed with SRP and/or Sec61 under conditions where there is no opportunity for elongation. Thus, it might be the case that upon initial SRP or Sec61 binding, a subtle pulling force would allow tRNA accommodation, after which the pause is relieved. Without any such tRNAs in the experimental system, the structure might simply reflect relaxation to the lowest energy state. Without carefully controlled pausing experiments in a translationally competent system with and without SRP and/or Sec61, their roles in relieving pausing cannot be deduced. The experimental evidence merely indicates that neither SRP nor Sec61 stably and quantitatively pull the nascent chain out of its arrested state in this experimental system. in vivo however, it might be that a transient tug (e.g., of HR2 engaging the Sec61 lateral gate) is enough to relieve the pause, but such a weak or dynamic HR2-Sec61 interaction is not captured in structure. Thus, the conclusions about the possible roles of SRP or Sec61 should be qualified, with alternatives such as the above included for completeness.

2) There is considerable discussion about the UPR in both the Introduction and Discussion. However, the study is mostly about a structure-function analysis of a stalling sequence, so an introduction providing a reader sufficient background on translation pausing and stalling mechanisms would be more appropriate. We believe that a change in emphasis of the Introduction and Discussion away from the UPR and more toward stalling mechanisms to be a very important.

3) The authors state that they were "surprised" to learn that the XBP1u AP has not evolved for maximal stalling. However, it was already shown some time ago that the AP can be mutated to enhance stalling, and in fact, they use such a mutant for their structures. Thus, the conclusion that the AP has not evolved for maximal stalling is neither new nor surprising. More generally, we do not feel the authors can draw conclusions on the evolution of the AP without information about whether the stall strength is conserved across species whose APs have variant sequences. The authors should re-write these sections accordingly.

4) All figures should clearly state what map is displayed and a σ value if possible. This will allow a reader to better judge how convincing the structural results are. For example, does Figure 3 depict a 2Fo-Fc map, or the sharpened EM density map? The former would be a less convincing result than the latter.

5) The mechanism of translational arrest would be significantly more convincing if the authors provided evidence that their data directly supports the relatively small changes they report to nucleotides in the PTC. A supplemental figure would be helpful that depicts the density for these residues in a sharpened EM density map.

6) Given the author's extensive mutational analysis, there is only a brief discussion about how the mutational data support the proposed model for the conformation of the nascent chain in the exit tunnel. It seems the authors have missed an opportunity to elaborate on how residues involved in either the turn or specific hydrogen bonding interactions with the ribosome or the nascent chain itself support their model. The authors could comment on whether the mutagenesis supports each of the nascent chain interactions described in the structural part of the paper.

7) The diagram in Figure 1 shows the HR2 sequence inserted into Sec61. The authors might wish to depict this differently to avoid implying that HR2 engages Sec61, which the findings suggest it does not do, at least not stably.

8) Figure 2 legend – "Traverse" should probably be "Transverse"; might be helpful to define N-tag and C-tag in the legend (or on the figure).

9) It is somewhat confusing to refer to different regions of the exit tunnel as 'upper' and 'lower,' as there is no convention in the field (see second paragraph of subsection “Generation and cryo-EM analysis of XBP1u-paused ribosome-nascent chain complexes”). Perhaps something like "proximal" or "distal" relative to the PTC?

10) Subsections “Generation and cryo-EM analysis of XBP1u-paused ribosome-nascent chain complexes” and “XBP1u nascent chain in the ribosomal tunnel”: No biochemical verification is provided to document that the rabbit ribosome would stall at the 'right' codon of XBP1u template in the constructs they used to prepare the complexes for cryo-EM reconstructions. Ideally, it would be good to provide biochemical evidence that the stalling takes place at the expected codon of the mRNA. Without this evidence the discussion of the complex, in particular the discussion of the fitting of the nascent chain sequence within the density, should be more tentative.

11) Subsection “PTC silencing by the XBP1u peptide”: The discussion of the induced/uninduced state of the PTC and the role of C4398 in the mechanism of stalling is confusing. The author say first that '80S ribosomes display the classical uninduced state of the PTC". However, few lines down we find that C2439 "is in the closed conformation, a position that.[…] is observed only after A-site accommodation". Furthermore, while clash of Asn261 with the incoming A-site tRNA makes sense as a part of the stalling mechanism, the pre-accommodation of C2498 is more questionable and in theory its 'pre-arrangement' could facilitate tRNA accommodation. Either the possible negative role of C2498 should be explained better or it should be left out from the proposed mechanism.

12) The placement of HR2 hydrophobic segment in the proximity to the SRP54 M-domain is drawn from the generally acceptable yet very indirect reasoning. Therefore the statement that "We. […] show that HR2 segment is recognized as a canonical signal sequence by SRP" is a clear overstatement.

13) Subsection “Saturation mutagenesis of the XBP1u pausing motif”: The mutagenesis experiments have been done in the context of the artificial construct where (likely) pulling force alleviates the arrest. As authors indicate in the Discussion, pulling force plays either no or very limited role in the normal operation of the XBP1u stalling system. Therefore, all the effects they observed with their mutations should be presented and discussed in the context of affecting sensitivity to the pulling force rather than as increasing or decreasing stalling efficiency of the XBP1u arrest peptide. Stalling efficiency and resistance of stalling to the pulling force might be related, or might not. The remedy would be to use an independent method, e.g. toeprinting, to demonstrate that the selected 'better' peptides exert more efficient stalling in the absence of the pulling force; but this has not been done. Thus, the authors should be more cautious in their interpretation.

14) The constructs used for cryo-EM complex preparation are not described in sufficient detail. The authors should present at least the amino acid sequences of the encoded protein(s) and ideally, also the nucleotide sequences of the constructs used.

15) Subsection “Interactions stabilizing the XBP1u peptide conformation”: In regards to interactions of W249 and W256 with the hydrophobic crevice in the tunnel, it is probably worthwhile mentioning that Martinez et al. (NAR 42, 11245) proposed that his crevice is involved in recognition of Trp during the TnaC-controlled translation arrest.

16) For the citation of Schmeing et al., 2005 in the Results remove "Martin"

17) Figure 2 legend. 'AP' segment is labeled as 'PS' in panel A.

18) Neither PKRM, nor RM are common acronyms. Define at the first mentioning.

19) Rephrase: 'mRNA was linearized'

20) Materials and methods section: "incubated with beads…" What beads? How much? And later in the section: if 'beads' should it then be supernatant, not 'flow-through'?

21) The authors do not include a complete list of linkers used in Figure 6B. This needs to be added as a supplementary table.

22) Figure 6 is hard to follow. Some descriptions are scattered in the text. For panel B, the authors should note the length of the SecM AP (since L is defined from the end of H to the end of the AP, and 20 aa does not make sense for XBP1u AP). Panels C and D axes are not labeled, and could also use a key for what 0 and 1 mean. The authors could also mark the positions that are already optimal in the WT sequence, in panel C, including a note of the 8 in the turn motif. This would help readers see the 6/8 mentioned in the text.

23) The authors detect a substantial fraction of the RNCs in the rotated state, suggesting that the AP structure perturbs translocation as well as likely preventing rapid A-site tRNA accommodation. Do the authors see any perturbation of the P-tRNA 3'-CCA end pairing with the P-loop in 28S rRNA, as was seen in the SecM-Pro case (https://elifesciences.org/articles/09684) and more recently with compounds that can stall translation (https://www.biorxiv.org/content/10.1101/315325v2)? The authors should comment on this point.

---

## [Author Response]

Overall, this is very interesting paper, the quality of the data is high, and it is generally well written. The main issues raised by the reviewers include over-interpretation of some of the findings, the need for a more complete description of some of the figures, and improved details in parts of the Materials and methods. All of these issues can be addressed without additional experimental work. The following issues should be addressed in preparing a revised manuscript:

*1) The authors' claims that SRP and Sec61 do not participates in relieving pausing is may lack strong experimental support and should therefore be qualified. In essence, they are looking at an artificial situation in which a purified stalled complex is mixed with SRP and/or Sec61 under conditions where there is no opportunity for elongation. Thus, it might be the case that upon initial SRP or Sec61 binding, a subtle pulling force would allow tRNA accommodation, after which the pause is relieved. Without any such tRNAs in the experimental system, the structure might simply reflect relaxation to the lowest energy state. Without carefully controlled pausing experiments in a translationally competent system with and without SRP and/or Sec61, their roles in relieving pausing cannot be deduced. The experimental evidence merely indicates that neither SRP nor Sec61 stably and quantitatively pull the nascent chain out of its arrested state in this experimental system.* in vivo *however, it might be that a transient tug (e.g., of HR2 engaging the Sec61 lateral gate) is enough to relieve the pause, but such a weak or dynamic HR2-Sec61 interaction is not captured in structure. Thus, the conclusions about the possible roles of SRP or Sec61 should be qualified, with alternatives such as the above included for completeness.*

We thank the reviewers for this insightful comment. Our findings, that the nascent chain states are unaltered during ER targeting are based on these in vitroreconstitution experiments; accordingly we have added a few sentences in the Results and Discussion mentioning the possible alternative scenarios in vivo, as suggested.

2) There is considerable discussion about the UPR in both the Introduction and Discussion. However, the study is mostly about a structure-function analysis of a stalling sequence, so an introduction providing a reader sufficient background on translation pausing and stalling mechanisms would be more appropriate. We believe that a change in emphasis of the Introduction and Discussion away from the UPR and more toward stalling mechanisms to be a very important.

We agree with this assessment and changed the emphasis of the Introduction and the Discussion accordingly. Now, the Introduction provides a broader overview of the general features of stalling APs. In the Discussion we explain in more detail to what end the XBP1u AP fits into the general scheme of APs and what its unique features with respect to stalling its mechanism are.

3) The authors state that they were "surprised" to learn that the XBP1u AP has not evolved for maximal stalling. However, it was already shown some time ago that the AP can be mutated to enhance stalling, and in fact, they use such a mutant for their structures. Thus, the conclusion that the AP has not evolved for maximal stalling is neither new nor surprising. More generally, we do not feel the authors can draw conclusions on the evolution of the AP without information about whether the stall strength is conserved across species whose APs have variant sequences. The authors should re-write these sections accordingly.

Point well taken. We have included a short section on the evolutionary conservation of the XBP1u AP (new Figure 6E), showing that the AP seems to be under selective pressure to maintain arather weak translational arrest efficiency.

4) All figures should clearly state what map is displayed and a σ value if possible. This will allow a reader to better judge how convincing the structural results are. For example, does Figure 3 depict a 2Fo-Fc map, or the sharpened EM density map? The former would be a less convincing result than the latter.

As suggested, we added σ values whenever possible, and to answer the reviewer’s question: Figure 3 depicts the (more convincing) sharpened EM density, which is clearly indicated in the Figure legend now.

5) The mechanism of translational arrest would be significantly more convincing if the authors provided evidence that their data directly supports the relatively small changes they report to nucleotides in the PTC. A supplemental figure would be helpful that depicts the density for these residues in a sharpened EM density map.

As suggested, we have a new Figure 4—figure supplement 1, showing the density for this critical base.

6) Given the author's extensive mutational analysis, there is only a brief discussion about how the mutational data support the proposed model for the conformation of the nascent chain in the exit tunnel. It seems the authors have missed an opportunity to elaborate on how residues involved in either the turn or specific hydrogen bonding interactions with the ribosome or the nascent chain itself support their model. The authors could comment on whether the mutagenesis supports each of the nascent chain interactions described in the structural part of the paper.

As suggested, we have added more detailed comments along these lines in the text.

7) The diagram in Figure 1 shows the HR2 sequence inserted into Sec61. The authors might wish to depict this differently to avoid implying that HR2 engages Sec61, which the findings suggest it does not do, at least not stably.

As suggested, we have a new, slightly changed Figure 1, with the HR2 domain partially but not fully engaging Sec61.

8) Figure 2 legend – "Traverse" should probably be "Transverse"; might be helpful to define N-tag and C-tag in the legend (or on the figure).

As suggested, we describe now the details of the N- and C-terminal tags of our construct used for RNC purification (Figure 2 legend). We also changed ‘Traverse’ to ‘Transverse’.

9) It is somewhat confusing to refer to different regions of the exit tunnel as 'upper' and 'lower,' as there is no convention in the field (see second paragraph of subsection “Generation and cryo-EM analysis of XBP1u-paused ribosome-nascent chain complexes”). Perhaps something like "proximal" or "distal" relative to the PTC?

We agree and have changed the text exactly as suggested.

10) Subsections “Generation and cryo-EM analysis of XBP1u-paused ribosome-nascent chain complexes” and “XBP1u nascent chain in the ribosomal tunnel”: No biochemical verification is provided to document that the rabbit ribosome would stall at the 'right' codon of XBP1u template in the constructs they used to prepare the complexes for cryo-EM reconstructions. Ideally, it would be good to provide biochemical evidence that the stalling takes place at the expected codon of the mRNA. Without this evidence the discussion of the complex, in particular the discussion of the fitting of the nascent chain sequence within the density, should be more tentative.

We indeed have not provided any additional biochemical evidence that the XBP1u stalling occurs at the expected site in our system. However, there is clear evidence in the literature that a unique stall site with Met260 as the most C-terminal amino acid is employed by the highly conserved nascent XBP1u in mammalian cells (Ingolia et al., 2011). Moreover, the structural detail of our reconstruction is resolved for the relevant part of the nascent XBP1u chain between 3.0 and 3.5 Å (Figure 2—figure supplement 3). This level of detail allowed for an unambiguous building of the molecular model (Figure 3), in particular with several bulky amino acid side chains (M260, K257, W256, H252, etc.) serving as clear landmarks for the correctness of the building process. It would not be possible to build a nascent chain model into the density after changing the register by +/- 1 or more amino acids. The identity of the stall site is further confirmed by the perfectly matching density for the codon-anticodon helix and for the P-site tRNA representing AUG:methionyl-tRNA. Therefore, we feel confident that our fitting of the nascent chain sequence within the density is highly reliable and that additional biochemical evidence corroborating the stall site is dispensable. For these reasons we would like to leave the discussion regarding fitting of the nascent chain into the density essentially as it is and suggest to not use a more tentative wording.

11) Subsection “PTC silencing by the XBP1u peptide”: The discussion of the induced/uninduced state of the PTC and the role of C4398 in the mechanism of stalling is confusing. The author say first that '80S ribosomes display the classical uninduced state of the PTC". However, few lines down we find that C2439 "is in the closed conformation, a position that.[…] is observed only after A-site accommodation". Furthermore, while clash of Asn261 with the incoming A-site tRNA makes sense as a part of the stalling mechanism, the pre-accommodation of C2498 is more questionable and in theory its 'pre-arrangement' could facilitate tRNA accommodation. Either the possible negative role of C2498 should be explained better or it should be left out from the proposed mechanism.

Apparently, the explanation regarding the conformations of the different bases was somewhat confusing as pointed out. In the revised text we made an effort to more clearly explain that on one hand the often perturbed U4531 (U2585) appears in our case in its normal uninduced conformation, whereas on the other hand, the C4398 (C2452) is stabilized by Leu259 in a prematurely closed or induced conformation, thereby together contributing to stalling by preventing A-site tRNA accommodation.

12) The placement of HR2 hydrophobic segment in the proximity to the SRP54 M-domain is drawn from the generally acceptable yet very indirect reasoning. Therefore the statement that "We.[…] show that HR2 segment is recognized as a canonical signal sequence by SRP" is a clear overstatement.

As suggested we re-phrased this sentence in order to account for the lack of direct evidence. It reads now: ‘…HR2 segment of XBP1u is likely to be recognized as…’

13) Subsection “Saturation mutagenesis of the XBP1u pausing motif”: The mutagenesis experiments have been done in the context of the artificial construct where (likely) pulling force alleviates the arrest. As authors indicate in the Discussion, pulling force plays either no or very limited role in the normal operation of the XBP1u stalling system. Therefore, all the effects they observed with their mutations should be presented and discussed in the context of affecting sensitivity to the pulling force rather than as increasing or decreasing stalling efficiency of the XBP1u arrest peptide. Stalling efficiency and resistance of stalling to the pulling force might be related, or might not. The remedy would be to use an independent method, e.g. toeprinting, to demonstrate that the selected 'better' peptides exert more efficient stalling in the absence of the pulling force; but this has not been done. Thus, the authors should be more cautious in their interpretation.

It is true that the mutagenesis screen was done under high pulling force, in contrast to the situation when the XBP1 AP performs its normal function in vivo. However, Figure 6B shows that a mutation (P254C) that increases the stalling strength at high pulling force does so also under low or negligible pulling force, arguing that stalling efficiency in the absence of pulling forces and resistance of stalling to the pulling force in fact are related and co-vary. We now address this issue in the text, and have also made it more clear that the data in Figure 6C and D were obtained under high pulling force.

14) The constructs used for cryo-EM complex preparation are not described in sufficient detail. The authors should present at least the amino acid sequences of the encoded protein(s) and ideally, also the nucleotide sequences of the constructs used.

As suggested, we have added both the nucleic acid and amino acid sequence of the construct in the Materials and methods section.

15) Subsection “Interactions stabilizing the XBP1u peptide conformation”: In regards to interactions of W249 and W256 with the hydrophobic crevice in the tunnel, it is probably worthwhile mentioning that Martinez et al. (NAR 42, 11245) proposed that his crevice is involved in recognition of Trp during the TnaC-controlled translation arrest.

As suggested, we included a sentence and cited the paper accordingly.

16) For the citation of Schmeing et al., 2005 in the Results remove "Martin"

“Martin” has been removed.

17) Figure 2 legend. 'AP' segment is labeled as 'PS' in panel A.

It was initially described as a pausing sequence (PS). We have now changed it to ‘AP’ in order to be consistent through the manuscript.

18) Neither PKRM, nor RM are common acronyms. Define at the first mentioning.

Thanks for pointing this out; we defined now at the first mentioning.

19) Rephrase: 'mRNA was linearized'

We have rephrased (and corrected) it to ‘mRNA was denatured’.

20) Materials and methods section: "incubated with beads…" What beads? How much? And later in the section: if 'beads' should it then be supernatant, not 'flow-through'?

Thank you for pointing out this lack of detailed information. We added now more information regarding the purification, and also incorporated the suggested changes.

21) The authors do not include a complete list of linkers used in Figure 6B. This needs to be added as a supplementary table.

As suggested, we added this information in a Supplementary File 2.

22) Figure 6 is hard to follow. Some descriptions are scattered in the text. For panel B, the authors should note the length of the SecM AP (since L is defined from the end of H to the end of the AP, and 20 aa does not make sense for XBP1u AP). Panels C and D axes are not labeled, and could also use a key for what 0 and 1 mean. The authors could also mark the positions that are already optimal in the WT sequence, in panel C, including a note of the 8 in the turn motif. This would help readers see the 6/8 mentioned in the text.

We have amended the figure as suggested.

23) The authors detect a substantial fraction of the RNCs in the rotated state, suggesting that the AP structure perturbs translocation as well as likely preventing rapid A-site tRNA accommodation. Do the authors see any perturbation of the P-tRNA 3'-CCA end pairing with the P-loop in 28S rRNA, as was seen in the SecM-Pro case (https://elifesciences.org/articles/09684) and more recently with compounds that can stall translation (https://www.biorxiv.org/content/10.1101/315325v2)? The authors should comment on this point.

We checked this point again carefully and did not find any perturbations of the tRNA 3'-CCA end pairing with the P-loop in 28S rRNA. As suggested, we comment on this issue in the revised text in line numbers (subsection “PTC silencing by the XBP1u peptide”).